# NATIVE REASONING MODELS: TRAINING LANGUAGE MODELS TO REASON ON UNVERIFIABLE DATA

**Yuanfu Wang[1]**    **Zhixuan Liu[1,2]**    **Xiangtian Li[1]**    **Chaochao Lu[1]**    **Chao Yang[1]**[*]
[1]Shanghai Artificial Intelligence Laboratory    [2]Shanghai Jiao Tong University
sharkwyf@gmail.com, lzx993124494@sjtu.edu.cn
{lixiangtian, luchaochao, yangchao}@pjlab.org.cn

## ABSTRACT

The prevailing paradigm for training large reasoning models—combining Supervised Fine-Tuning (SFT) with Reinforcement Learning with Verifiable Rewards (RLVR)—is fundamentally constrained by its reliance on high-quality, human-annotated reasoning data and external verifiers. This dependency incurs significant data-collection costs, risks embedding human cognitive biases, and confines the reinforcement learning stage to objectively assessable domains like mathematics and coding, leaving a wide range of unverifiable tasks beyond its scope. To overcome these limitations, we introduce **NRT** (**N**ative **R**easoning **T**raining), a novel framework that cultivates complex reasoning by having the model generate its own reasoning traces using only standard question-answer pairs, thereby obviating the need for expert-written demonstrations. NRT reframes the training problem by treating the reasoning process as a latent variable. It employs a unified training objective that models reasoning as an optimization problem, intrinsically rewarding paths that increase the model's likelihood of producing the ground-truth answer. This unified perspective allows us to analyze intrinsic failure modes of prior methods, such as policy collapse, and systematically design more robust reward aggregation functions, creating a self-reinforcing feedback loop where the model learns to *think* in ways that resolve its own uncertainty. Empirical evaluation on Llama and Mistral model families demonstrates that NRT achieves state-of-the-art performance among verifier-free methods, significantly outperforming standard SFT baselines and prior verifier-free RL methods. Our approach yields particularly strong performance gains in complex reasoning domains and exhibits high robustness to policy collapse, offering a general, scalable path toward building more powerful and broadly applicable reasoning systems.

## 1 INTRODUCTION

Large Language Models (LLMs) have evolved into powerful Large Reasoning Models (LRMs), demonstrating remarkable abilities in complex tasks like mathematics, coding, and multi-step problem-solving. The prevailing approach is a two-stage pipeline: Supervised Fine-Tuning (SFT) followed by Reinforcement Learning with Verifiable Rewards (RLVR) (DeepSeek-AI et al., 2025). In the first stage, SFT trains the model to imitate human-written reasoning processes, often as chain-of-thought demonstrations (Wei et al., 2022), a *cold-start* phase that provides a foundational ability to structure its thoughts. The second stage, RLVR, refines and generalizes this ability, using an external verifier—such as a unit test for code or a final answer check for a math problem—to reward the model for any reasoning trace yielding a correct output, freeing it from strictly mimicking a single human-provided path (Shao et al., 2024).

Despite its success, the SFT+RLVR paradigm has two fundamental limitations. First, it depends critically on the initial SFT phase, which requires expensive, high-quality, human-annotated reasoning data. This data can embed human priors and biases, constraining the model's performance and its search for novel, more effective reasoning strategies. Second, reliance on an external verifier confines the RLVR stage to objectively assessable domains. This leaves a vast landscape of tasks—such

---

[*]Corresponding author.

as open-ended question answering, creative writing, and summarization—where reasoning is crucial but correctness is subjective and unverifiable. Attempts to bridge this gap often use proxy rewards. Some methods use a judge model to score outputs (Vacareanu et al., 2024; Guan et al., 2024), but this introduces rubric design complexity and is constrained by the judge's own capabilities. Other self-rewarding approaches Chen et al. (2024) train a model to generate its own feedback but can be limited by the model's initial capacity and often suffer mode collapse, where the policy converges to simple, low-entropy outputs. These limitations motivate a fundamental question: *how can we cultivate and refine reasoning abilities in language models using only question-answer pairs, especially for domains where external verifiers are unavailable?*

In this work, we introduce *Native Reasoning Training (NRT)*, a novel framework to cultivate reasoning abilities without supervision on the reasoning trace itself. The term *Native* signifies that the reasoning process is not imitated from human demonstrations but emerges as the model discovers how to connect a question to its answer. It treats the reasoning process as a latent variable to be discovered, not imitated. NRT uses RL to explore potential reasoning traces, intrinsically rewarding those that increase the model's likelihood of generating the ground-truth answer. This creates a self-reinforcing feedback loop where the model learns to generate reasoning traces that resolve its own uncertainty. Crucially, this process requires only standard question-answer pairs, obviating the need for expert-written reasoning demonstrations and external verifiers. NRT unlocks reasoning training for a vast range of problems where verifiers are unavailable.

Our contributions are threefold: (1) **A Unified Framework for Verifier-Free Reasoning.** We introduce NRT, a unified framework that models reasoning as a latent variable and optimizes the marginal log-likelihood of the answer. Our framework contextualizes prior methods, enabling a theoretical analysis of their reward structures and failure modes like policy collapse. (2) **A Powerful, Principled Training Method for Unverifiable Data.** We propose a principled training method with novel reward schemes (e.g., NRT-WS) derived from our framework. This method enhances reasoning by focusing on the model's uncertainty, requires only question-answer pairs, and eliminates the need for expert demonstrations or external verifiers, thus lowering data costs and broadening applicability. (3) **State-of-the-Art Performance through Refined Reasoning.** We demonstrate empirically that NRT achieves state-of-the-art results, significantly outperforming SFT baselines and prior verifier-free methods on diverse reasoning benchmarks. Our analysis confirms that NRT is highly robust to training instabilities like policy collapse that affect other RL methods.

## 2 RELATED WORK

Our work on developing reasoning in LLMs builds on three paradigms:

**Reinforcement Learning with Verifiable Rewards (RLVR).** The predominant approach for training state-of-the-art reasoning models is RLVR, which requires an objective, external verifier to confirm the correctness of a model's final answer. While highly effective, this paradigm is largely confined to domains like mathematics and code generation where ground-truth can be programmatically checked (Shao et al., 2024; DeepSeek-AI et al., 2025; OpenAI et al., 2024). The foundational method is Group Relative Policy Optimization (GRPO) (Shao et al., 2024), built on general policy optimization algorithms like PPO (Schulman et al., 2017b). Its success spawned algorithmic refinements that improve optimization stability (Liu et al., 2025; MiniMax et al., 2025; Zheng et al., 2025) or adapt to problem difficulty (Yu et al., 2025a). Despite these advances, the core reliance on external verifiers constrains these powerful RL techniques to objectively assessable problem domains.

**Proxy Rewards and Self-Rewarding.** To extend reasoning training to more general domains, researchers developed proxy reward methods. One strategy uses a powerful LLM judge to score outputs against a predefined rubric or human preferences (Lee et al., 2023; Guan et al., 2024; Yuan et al., 2024). However, this approach introduces rubric design complexity and is bottlenecked by the judge's own capabilities and biases (Gao et al., 2023; Lightman et al., 2023). Another direction leverages self-reward, where the model generates its own feedback. These techniques often reward high-confidence or consistent answers, identified via mechanisms like majority voting or entropy minimization (Zuo et al., 2025; Cheng et al., 2025). These objectives, while innovative, can penalize exploration, reduce response diversity, and reinforce the model's initial biases (Cui et al., 2025).

**Reinforcement Learning on Unverifiable Data.** The most related work trains reasoning directly on non-verifiable data, dispensing with external verifiers and proxy judges. These methods pioneer using the ground-truth answer itself as the core reward signal. One approach treats the reasoning trace as a latent variable and optimizes for internal steps that maximize the likelihood of the correct final answer, often using the evidence lower bound or related objectives (Tang et al., 2025; Zhou et al., 2025). Another emerging class uses the model's predictive confidence in the final answer as an intrinsic reward to guide policy updates (Yu et al., 2025b; Chen et al., 2024). Recently, Direct Reasoning Optimization (DRO) introduced a reasoning reflection reward that identifies salient tokens in a reference answer to guide training on open-ended tasks (Xu et al., 2025). Our work, Native Reasoning Training (NRT), falls within this scope. It advances this paradigm with a unified framework that formalizes these intrinsic reward mechanisms and can cultivate complex reasoning from a base model using only question-answer pairs, without requiring reasoning demonstrations.

## 3 METHOD

We first review Supervised Fine-Tuning (SFT) and Reinforcement Learning with Verifiable Rewards (RLVR), to contextualize our approach. Then we detail the training objective, optimization process, reward shaping, and mechanisms for enforcing structured reasoning trace of our proposed method.

### 3.1 PRELIMINARIES: STANDARD PARADIGMS FOR REASONING TRAINING

**Supervised Fine-Tuning (SFT).** SFT adapts a pre-trained model $\pi_\theta$ to mimic expert demonstrations from a dataset $\mathcal{D}$, where each instance is a triplet $(x, z^\star, y^\star)$ of an input question, an expert reasoning trace, and a final answer. The objective is to maximize the conditional log-likelihood of the expert sequence $o^\star = (z^\star, y^\star)$, expressed as:

$$J_{\text{SFT}}(\theta) = \mathbb{E}_{(x,z^\star,y^\star) \sim \mathcal{D}} \left[ \log \pi_\theta(z^\star, y^\star | x) \right]. \tag{1}$$

As language models are auto-regressive, this log-likelihood over the token sequence $o^\star = (t_1, \ldots, t_N)$ is decomposed as:

$$\log \pi_\theta(o^\star | x) = \sum_{i=1}^{N} \log \pi_\theta(t_i | x, t_{<i}). \tag{2}$$

This phase instills a strong prior for reasoning structure, preparing it for subsequent optimization.

**Reinforcement Learning with Verifiable Rewards (RLVR).** Unlike imitation-based SFT, Reinforcement Learning (RL) optimizes the policy $\pi_\theta$ to maximize an expected reward $R(x, o)$ for a generated output $o = (z, y)$. The general objective is:

$$\max_\theta \mathbb{E}_{o \sim \pi_\theta(\cdot | x)}[R(x, o)]. \tag{3}$$

In RLVR, the reward is determined by an external verifier assessing the correctness of the final answer $y$ against the ground truth $y^\star$. The reward is typically binary:

$$R_{\text{Verifier}}(y; y^\star) = \mathbf{1}_{\{y \equiv y^\star\}}, \tag{4}$$

where $\mathbf{1}_{\{\cdot\}}$ is the indicator function. The RLVR objective is thus:

$$J_{\text{RLVR}}(\theta; x, y^\star) = \mathbb{E}_{z \sim \pi_\theta(\cdot | x)} \left[ \mathbb{E}_{y \sim \pi_\theta(\cdot | x, z)}[R_{\text{Verifier}}(y; y^\star)] \right]. \tag{5}$$

This objective, optimized via policy gradient algorithms like PPO (Schulman et al., 2017a) or GRPO (Shao et al., 2024), reinforces any reasoning trace $z$ that yields a correct answer. The efficacy of RLVR is contingent on a reliable verifier, confining its application to domains like mathematics and code generation.

### 3.2 NATIVE REASONING TRAINING

Native Reasoning Training (NRT) cultivates reasoning without explicit supervision on the reasoning trace, requiring only the input question $x$ and the reference answer $y^\star$. The core idea is to treat the

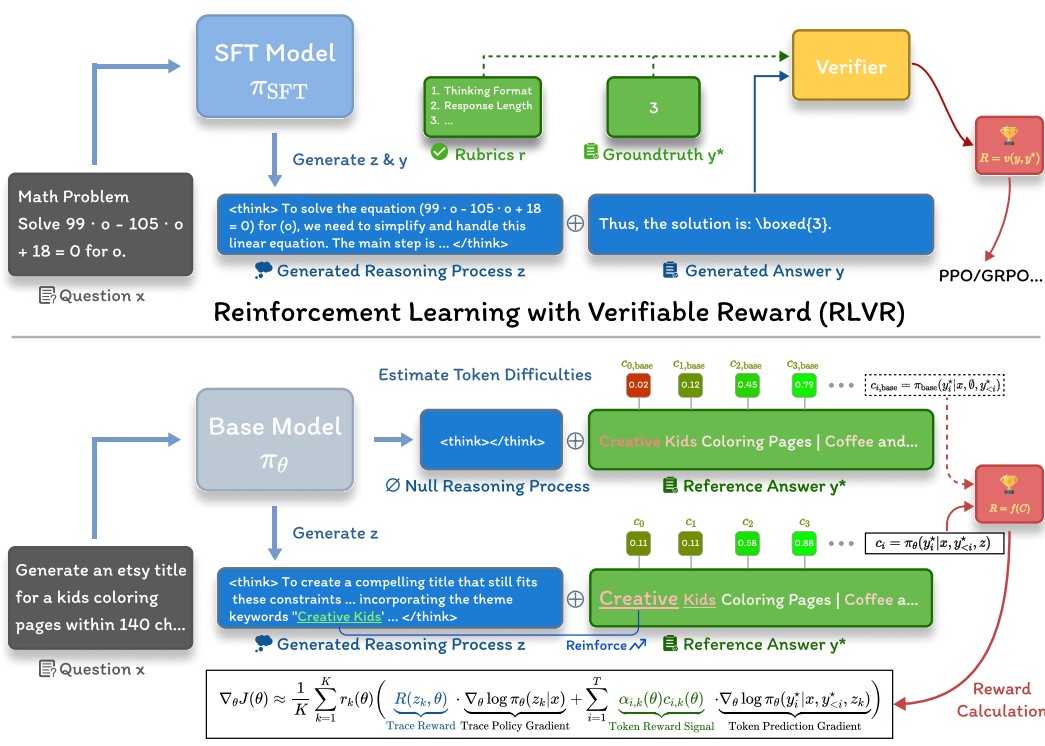

Figure 1: Comparison of Reinforcement Learning with Verifiable Rewards (RLVR) and our Native Reasoning Training (NRT). **(Top)** RLVR uses an external verifier to reward reasoning $z$ that yields an answer $y$ matching the ground-truth $y^\star$. This approach is constrained by its need for a verifier. **(Bottom)** NRT operates on general SFT data, using only a question $x$ and a reference answer $y^\star$. It trains the model to generate a latent reasoning trace $z$ by intrinsically rewarding traces that increase its own predictive confidence in the reference answer. This self-reinforcing process removes the need for external verifiers or expert-written reasoning.

reasoning trace $z$ as a latent variable and intrinsically reward traces that increase the model's own predictive confidence in the reference answer.

We formalize this principle at the token level. Let the reference answer be $y^\star = (y_1^\star, \ldots, y_T^\star)$. A per-trace reward, $R(z, \theta)$, is defined by applying a differentiable aggregation function $f : \mathbb{R}^T \to \mathbb{R}$ to the model's token-level conditional probabilities $c_i(z, \theta) = \pi_\theta(y_i^\star | x, z, y_{<i}^\star)$:

$$R(z, \theta) = f\big(c_1(z, \theta), c_2(z, \theta), \ldots, c_T(z, \theta)\big). \tag{6}$$

The NRT objective is the expectation of this reward, encouraging the generation of helpful traces:

$$J(\theta) = \mathbb{E}_{z \sim \pi_\theta(z|x)}\big[R(z, \theta)\big]. \tag{7}$$

Maximizing this objective teaches the model to generate reasoning that is helpful for its token-by-token predictive process. We optimize this objective using off-policy reinforcement learning. As derived in Appendix B, the gradient can be estimated from a batch of traces $\{z_k\}_{k=1}^{K}$ sampled from an older policy $\pi_{\text{old}}$:

$$\nabla_\theta J(\theta) \approx \frac{1}{K} \sum_{k=1}^{K} r_k(\theta) \bigg( \underbrace{R(z_k, \theta)}_{\text{Trace Reward}} \cdot \underbrace{\nabla_\theta \log \pi_\theta(z_k|x)}_{\text{Trace Policy Gradient}}$$
$$+ \sum_{i=1}^{T} \underbrace{\alpha_{i,k}(\theta) c_{i,k}(\theta)}_{\text{Token Reward Signal}} \cdot \underbrace{\nabla_\theta \log \pi_\theta(y_i^\star | x, y_{<i}^\star, z_k)}_{\text{Token Prediction Gradient}} \bigg), \tag{8}$$

where $r_k(\theta)$ is the importance sampling ratio, $c_{i,k}(\theta)$ is the conditional probability of token $y_i^\star$, and $\alpha_{i,k}(\theta) = \frac{\partial f}{\partial c_i}$ is the reward function's sensitivity to $c_{i,k}(\theta)$.

This gradient provides two complementary learning signals. The first term is a policy gradient update reinforcing holistically useful traces ($z_k$) via the overall **Trace Reward**. The second is a weighted, per-token policy gradient for answer prediction, which updates the model to better predict answer tokens ($y_i^\star$) given the trace ($z_k$), weighting each token's gradient by the **Token Reward Signal**. This dual mechanism jointly refines the trace generation and answer prediction policies, enabling self-correction of the latent reasoning process without external trace supervision.

### 3.2.1 INTRINSIC REWARD SHAPING VIA AGGREGATION FUNCTIONS

The choice of differentiable aggregation function $f$ is central to NRT, shaping the intrinsic reward landscape by defining two key gradient components in Equation 8: the overall Trace Reward $R(z,\theta) = f(\mathbf{c})$, which scales the trace policy gradient; and the Token Reward Signal $\alpha_i c_i = \frac{\partial f}{\partial c_i} c_i$, which weights each answer token's policy gradient.

Different forms of $f$ prioritize different aspects of the prediction task, such as holistic accuracy or improving the least-confident predictions. Table 1 summarizes several aggregation functions and their resulting gradient components, derived in Appendix C.

Table 1: Intrinsic reward shaping schemes derived from different aggregation functions $f$. $c_i = \pi_\theta(y_i^\star|x, z, y_{<i}^\star)$ is the conditional probability of the $i$-th token of the reference answer.

| Scheme | Aggregation Function $f(\mathbf{c})$ | Trace Reward $R(z,\theta)$ | Token Reward Signal $\frac{\partial f}{\partial c_i} c_i$ |
|---|---|---|---|
| Sequence Log-Prob (logP) | $f(\mathbf{c}) = \sum_{j=1}^{T} \log c_j$ | $\log \pi_\theta(y^\star|x, z)$ | 1 |
| Sequence Probability (P) | $f(\mathbf{c}) = \prod_{j=1}^{T} c_j$ | $\pi_\theta(y^\star|x, z)$ | $\pi_\theta(y^\star|x, z)$ |
| Geometric Mean (GM) | $f(\mathbf{c}) = \left(\prod_{j=1}^{T} c_j\right)^{1/T}$ | $(\pi_\theta(y^\star|x, z))^{1/T}$ | $\frac{1}{T}(\pi_\theta(y^\star|x, z))^{1/T}$ |
| Arithmetic Mean (AM) | $f(\mathbf{c}) = \frac{1}{T}\sum_{j=1}^{T} c_j$ | $\frac{1}{T}\sum_{j=1}^{T} c_j$ | $\frac{1}{T} c_i$ |
| Weighted Sum (WS) | $f(\mathbf{c}) = \sum_{j=1}^{T} w_j c_j$ | $\sum_{j=1}^{T} w_j c_j$ | $w_i c_i$ |

The Sequence Log-Probability and Arithmetic Mean (AM) schemes correspond to prior work, but our unified framework reveals their potential pitfalls. AM, for instance, can be dominated by high-probability (easy) tokens, rewarding trivial or null traces ($z$) that fail to help with difficult tokens, which can cause the policy collapse observed in our experiments (Sec 4.3).

This motivates using more robust functions like Geometric Mean (GM). The GM scheme is more robust than AM because a single near-zero probability token collapses the entire reward, forcing the model to perform well on all tokens.

The Weighted Sum (WS) scheme offers more granular control by targeting points of highest uncertainty. A powerful heuristic is to focus on difficult tokens, measured by their low probability $c_{i,\text{base}} = \pi_{\text{base}}(y_i^\star|x, \emptyset, y_{<i}^\star)$ from a baseline model $\pi_{\text{base}}$, e.g. $\pi_{\text{old}}$, with a null trace ($z = \emptyset$). Setting weights inversely proportional to these probabilities encourages reasoning that resolves the model's own uncertainty. Table 2 details two such implementations.

Table 2: Practical implementations of the Weighted Sum scheme that prioritize difficult tokens. Weights $w_i$ are based on the model's baseline uncertainty, measured by $c_{i,\text{base}} = \pi_{\text{base}}(y_i^\star|x, \emptyset, y_{<i}^\star)$.

| Weighting Scheme | Weight Formula ($w_i$) | Trace Reward $R(z,\theta)$ | Token Reward Signal $\alpha_i c_i$ |
|---|---|---|---|
| Inverse-Probability ($1/p$) | $w_i \propto \frac{1}{c_{i,\text{base}}}$ | $\sum_{j=1}^{T} \frac{c_j}{c_{j,\text{base}}}$ | $\frac{c_i}{c_{i,\text{base}}}$ |
| Log-Probability ($-\log p$) | $w_i \propto -\log c_{i,\text{base}}$ | $-\sum_{j=1}^{T} c_j \log c_{j,\text{base}}$ | $-c_i \log c_{i,\text{base}}$ |

These targeted weighting strategies use the model's baseline uncertainty to shape rewards, creating a self-reinforcing loop that encourages reasoning to address its own predictive weaknesses.

### 3.2.2 REWARD STABILIZATION

To stabilize policy gradient updates, the reward in Equation 8 is replaced with a robust advantage estimate, $A_k(\theta)$, inspired by GRPO (Shao et al., 2024). For each prompt $x$, we generate a group of $K$ candidate traces $\{z_k\}_{k=1}^K$ and compute their relative advantages.

The advantage calculation is a two-step process. First, a clipped reward, $R'_k(\theta)$, is computed to focus the learning signal on traces that outperform a baseline. The baseline, $R_{\text{base}} = f(\mathbf{c}_{\text{base}})$, is the reward from an empty trace ($z = \emptyset$). The clipped reward for trace $z_k$ is:

$$R'_k(\theta) = \max\left(0, R(z_k, \theta) - R_{\text{base}}\right). \tag{9}$$

Second, these clipped rewards $\{R'_k\}_{k=1}^K$ are normalized to have zero mean and unit variance within their group. This yields the final advantage for trace $z_k$:

$$A_k(\theta) = \frac{R'_k(\theta) - \text{mean}(R')}{\text{std}(R')}, \tag{10}$$

where $\text{mean}(R')$ and $\text{std}(R')$ are the mean and standard deviation of the clipped rewards for the group. This group-wise normalization converts absolute rewards into a relative ranking, significantly reducing gradient variance and focusing optimization on identifying the most effective reasoning traces among candidates.

### 3.2.3 STRUCTURAL FORMAT SUPERVISION

We employ structural format supervision to teach the model to clearly distinguish the reasoning trace $z$ from the final answer $y$. This supervision guides the model to adopt a specific output format: a concatenation of (1) a start-of-reasoning token $t_{\text{start}}$, (2) the reasoning trace $z$, (3) an end-of-thought token $t_{\text{end}}$, and (4) the final answer $y$.

Instead of a rigid constraint, we apply a supplementary format-supervision loss, $L_{\text{format}}$. This is a targeted cross-entropy objective applied only to the special tokens:

$$L_{\text{format}}(\theta) = -\log \pi_\theta(t_{\text{start}}|x) - \log \pi_\theta(t_{\text{end}}|x, t_{\text{start}}, z). \tag{11}$$

This format loss encourages the model to generate well-formed structures without harshly penalizing exploration. To ensure all samples are structurally complete for the policy update, we manually append $t_{\text{end}}$ if it is missing at the maximum generation length, thereby maximizing data utility.

## 4 EXPERIMENTS

### 4.1 EXPERIMENTAL SETUP

**Models and Dataset.** We use the pretrained *Llama-3.2-3B*, *Mistral-7B-v0.3* and *Llama-3.1-8B* models (Dubois et al., 2024) for all experiments. All methods, including the SFT baseline and NRT variants, are trained on a shared 200K-sample random subset of the `tulu-3-sft-mixture` dataset (Lambert et al., 2024). The dataset contains only question-answer pairs $(x, y^\star)$, with no expert-written reasoning traces ($z^\star$). This setup ensures a fair comparison, as all methods start from the same un-tuned checkpoint and use identical data.

**Methods.** We evaluate our proposed Native Reasoning Training (NRT) methods against a comprehensive set of baselines. The compared methods are: (1) **SFT**, a standard Supervised Fine-Tuning model, our primary baseline. We also include our re-implementations of prior work, viewed as specific instances of our NRT framework (see Table 5 in Appendix D for a detailed comparison): (2) **JLB** (Tang et al., 2025), corresponding to our Sequence Log-Probability scheme, maximizing the log-probability of the entire answer sequence; (3) **Verifree** (Zhou et al., 2025), which aligns with the Sequence Probability scheme and maximizes the joint probability of the answer sequence; and (4) **RLPR** (Yu et al., 2025b), an instance of the Arithmetic Mean scheme that maximizes the average probability of individual answer tokens. Finally, we test three NRT variants based on our proposed reward shaping schemes: (5) **NRT-GM**, which uses the Geometric Mean to balance reward across all answer tokens; (6) **NRT-WS** $(1/p)$, a Weighted Sum scheme that prioritizes difficult tokens using their inverse baseline probability as weights; and (7) **NRT-WS** $(-\log p)$, a Weighted Sum scheme that uses the negative log baseline probability to focus on the most challenging tokens.

Table 3: Comprehensive evaluation across a suite of nine general and reasoning benchmarks. The comparison is conducted on Llama-3.2-3B, Mistral-7B-v0.3, and Llama-3.1-8B models, all fine-tuned on the same 200k subset of `tulu-3-sft-mixture`. NRT, particularly the weighted-sum variant, NRT-WS $(-\log p)$, demonstrates significant performance gains over all baselines. Bold and underline indicate the top two results per model. *Our baseline implementations.

| Method | General Reasoning | | Question Answering | | | Math | | Code | Instruct | Overall |
|---|---|---|---|---|---|---|---|---|---|---|
| | BBH | MMLU | DROP | PopQA | TQA | GSM8K | MATH | HEval | IFEval | |
| *Llama-3.2-3B-Base* | | | | | | | | | | |
| SFT | 30.2 | 44.3 | 24.6 | **23.3** | 46.2 | 31.7 | 8.5 | 55.2 | **46.3** | 36.4 |
| JLB* | 28.2 | 43.4 | 22.6 | 18.3 | 43.8 | 22.3 | 9.3 | 57.8 | 42.0 | 34.1 |
| Verifree* | 26.9 | 44.0 | 24.1 | 14.7 | 45.7 | 37.0 | 10.5 | 57.5 | 38.7 | 35.2 |
| RLPR* | 28.4 | 43.4 | 23.0 | 18.3 | 44.8 | 33.0 | 10.0 | 57.8 | 43.3 | 35.7 |
| NRT-GM | 29.4 | 43.5 | 23.9 | 18.3 | **47.5** | 46.0 | 9.1 | **61.8** | 39.3 | 37.3 |
| NRT-WS $(1/p)$ | 35.0 | 47.0 | **26.7** | 16.7 | 42.4 | 44.3 | 12.5 | 60.8 | 41.7 | 38.0 |
| NRT-WS $(-\log p)$ | **39.1** | **48.5** | 26.1 | 20.7 | 44.2 | **49.3** | **13.1** | 58.8 | 44.3 | **39.9** |
| *Mistral-7B-v0.3* | | | | | | | | | | |
| SFT | 36.7 | 53.6 | 28.6 | **23.7** | 47.4 | 30.0 | 12.5 | 66.3 | 61.0 | 40.0 |
| JLB* | 40.1 | 54.0 | 32.0 | 22.7 | 43.8 | 8.3 | 12.1 | 63.0 | 62.0 | 37.6 |
| Verifree* | 43.3 | 55.0 | 44.4 | 18.0 | 47.5 | 56.7 | 19.2 | 60.2 | 61.3 | 45.1 |
| RLPR* | 40.4 | 55.1 | 19.7 | **23.7** | 43.4 | **65.3** | 19.0 | 66.6 | 60.3 | 43.7 |
| NRT-GM | **44.3** | **58.3** | **46.6** | 20.7 | 48.3 | 57.0 | 20.7 | **68.4** | 59.0 | **47.0** |
| NRT-WS $(1/p)$ | 43.4 | 56.8 | 40.0 | 19.3 | **54.3** | 62.0 | 19.3 | 63.2 | 62.3 | 46.7 |
| NRT-WS $(-\log p)$ | 42.6 | 56.1 | 30.6 | 20.0 | 51.6 | 63.7 | **22.5** | 63.0 | **66.7** | 46.3 |
| *Llama-3.1-8B-Base* | | | | | | | | | | |
| SFT | 38.0 | 59.2 | 36.7 | 30.3 | 45.5 | 29.0 | 17.8 | 74.7 | 58.3 | 46.0 |
| JLB* | 40.2 | 59.0 | 36.1 | 31.0 | 46.9 | 54.0 | 19.0 | 74.3 | 58.0 | 49.0 |
| Verifree* | 35.7 | 58.3 | 33.5 | 27.7 | 46.6 | 54.3 | 19.4 | 76.3 | 59.3 | 48.1 |
| RLPR* | 41.2 | 58.7 | 32.5 | 29.3 | 45.6 | 65.0 | 27.8 | **77.8** | **61.3** | 50.8 |
| NRT-GM | **54.3** | 66.1 | 48.7 | 26.7 | **47.3** | 70.3 | **32.2** | 76.3 | 55.3 | 54.9 |
| NRT-WS $(1/p)$ | 50.3 | 65.1 | 34.5 | **32.7** | 46.8 | 66.3 | 29.1 | 75.9 | 60.7 | 53.3 |
| NRT-WS $(-\log p)$ | 51.0 | **66.7** | **52.2** | 31.0 | 46.1 | **76.0** | 30.7 | **77.8** | 59.0 | **56.2** |

**Evaluation.** We evaluate all models on nine diverse benchmarks, grouped into five categories: General Reasoning (BBH (Suzgun et al., 2022), MMLU (Hendrycks et al., 2020)), Question Answering (DROP (Dua et al., 2019), PopQA (Mallen et al., 2022), TruthfulQA (Lin et al., 2021)), Math (GSM8K (Cobbe et al., 2021), MATH (Hendrycks et al., 2021)), Code Generation (HumanEval (Chen et al., 2021)), and Instruction Following (IFEval (Zhou et al., 2023)). We report the primary metric for each benchmark (Table 3) and the average score across all nine to gauge overall performance. See Appendix G for further details.

**Implementation Details.** All RL methods are trained using GRPO (Shao et al., 2024) with a learning rate of 1e-5 and a batch size of 256. We generate reasoning traces up to 2048 tokens. A small format-supervision loss (weight 0.3) encourage the specified output format. See Appendix F.

## 4.2 MAIN RESULTS

**NRT establishes a new state-of-the-art for verifier-free reasoning, with particularly strong gains in complex, reasoning-intensive domains.** As shown in Table 3, its NRT methods achieve the highest average scores on Llama-3.2-3B, Mistral-7B-v0.3, and Llama-3.1-8B models. On the 8B model, our best variant, NRT-WS $(-\log p)$, scores 56.2 on average, a +10.2 point gain over the SFT model (46.0) and a +5.4 point improvement over the strongest prior method, RLPR (50.8). This trend holds for the 3B model, where NRT-WS $(-\log p)$ achieves a 39.9 average, outperforming the SFT baseline by 3.5 points. Notably, on the smaller 3B model, other verifier-free RL methods (JLB, Verifree, RLPR) fail to surpass the SFT baseline, highlighting both the task's difficulty and NRT's robustness. The benefits are most pronounced on challenging reasoning benchmarks. On GSM8K, NRT-WS $(-\log p)$ boosts the 8B model's score from 29.0 (SFT) to 76.0, far surpassing RLPR's 65.0. Similarly, NRT-GM improves the BBH score to 54.3, a substantial +13.1 points over RLPR,

demonstrating that NRT effectively elicits latent reasoning skills for complex, multi-step problems. For qualitative case studies illustrating these improvements, see Appendix I.

**NRT's success is driven by its principled reward shaping, which guides the model to resolve its own uncertainty and outperforms simpler aggregation methods.** While simpler variants like NRT-GM show significant improvement, the weighted-sum (WS) schemes consistently deliver the best results. These schemes validate the core hypothesis: guiding the model to generate reasoning that resolves its own uncertainty is a highly effective. By prioritizing tokens the model finds difficult (i.e., those with low baseline probability), NRT-WS $(-\log p)$ becomes the top-performing method overall for both model scales, securing the highest scores on GSM8K, MMLU, DROP, and HumanEval on the 8B model. Its superior performance over methods that use simpler aggregation functions—such as RLPR (Arithmetic Mean) and JLB (Sequence Log-Probability)—proves that the structure of the intrinsic reward is a key factor in the success of verifier-free learning.

### 4.3 Analysis of Training Dynamics of the Reasoning Process

**NRT effectively avoids the common RL pitfalls of mode collapse and policy degradation, maintaining diverse, lengthy, and high-quality reasoning throughout training.** A significant challenge in RL-based fine-tuning is mode collapse, where the policy converges to simple, low-reward outputs, ceasing to explore more complex strategies. We track the evolution of the reasoning process $z$ during training across three metrics: entropy, token length, and semantic quality (evaluated via an LLM judge; see Appendix H for details). As shown in Figure 2, the RLPR baseline exhibits a rapid degradation on both 3B and 8B models. It converges to a state producing short, low-entropy reasoning traces (Figures 2a-2d) that receive consistently low quality scores (Figures 2e-2f). This confirms that the baseline settles into a poor local optimum, generating repetitive or trivial content that fails to support logical deduction. In contrast, all NRT variants sustain high-entropy and long reasoning processes while maintaining or improving reasoning quality throughout training. This alignment between structural metrics (length, entropy) and semantic quality verifies that the NRT models are not merely "babbling" to increase length, but are actively engaging in substantive reasoning steps.

**NRT's superior training stability stems from a reward mechanism that incentivizes the generation of genuinely helpful reasoning over superficial shortcuts.** As analyzed in our framework (Sec 3.2.1), the RLPR baseline's collapse can be attributed to its Arithmetic Mean reward, which may inadvertently reward empty or trivial traces if they maximize the probability of a few easy-to-predict answer tokens. NRT-GM, through its geometric mean, penalizes any single token with a low probability, requiring the model to generate holistically helpful reasoning. More powerfully, NRT-WS schemes force the model to confront its weaknesses by up-weighting difficult tokens. To earn a high reward, the model must conduct complex, multi-step reasoning to resolve its most significant predictive uncertainties. This mechanism prevents the quality degradation observed in the baseline and translates directly to the superior downstream performance in Table 3.

### 4.4 Analysis of Groundtruth Token Probabilities

**NRT's targeted reward shaping effectively teaches the model to resolve its own predictive weaknesses, significantly boosting confidence on the most difficult tokens.** To understand how NRT improves the model's predictive capabilities, we analyze the confidence change for each ground-truth answer token. We categorize tokens by difficulty, measured by the SFT model's prediction entropy ($H_{\text{SFT}}$). As Figure 3a shows, this difficulty distribution is highly skewed: while most tokens are *easy* for the baseline to predict (low entropy, median 0.125), a long tail of highly uncertain, *hard* tokens exists. These hard tokens are precisely where reasoning is most critical.

**Weighted-sum NRT schemes uniquely focus learning on these hard-to-predict tokens, where other methods fail or degrade performance.** Figure 3b plots the relative improvement in token probability ($P/P_{\text{SFT}}$) against the token's baseline entropy, where a ratio ¿ 1 indicates improved confidence over the SFT baseline. The results reveal a clear divergence: RLPR and NRT-GM methods show negligible gains, with confidence hovering around or even dipping below the baseline for high-entropy tokens. In contrast, both weighted-sum (WS) variants exhibit a strong positive correlation. For the highest-entropy tokens, NRT-WS $(1/p)$ increases the assigned probability by up to 15%, providing direct evidence that our targeted reward shaping works as designed, forcing the model to generate reasoning that addresses its most critical points of confusion. By focusing the learn-

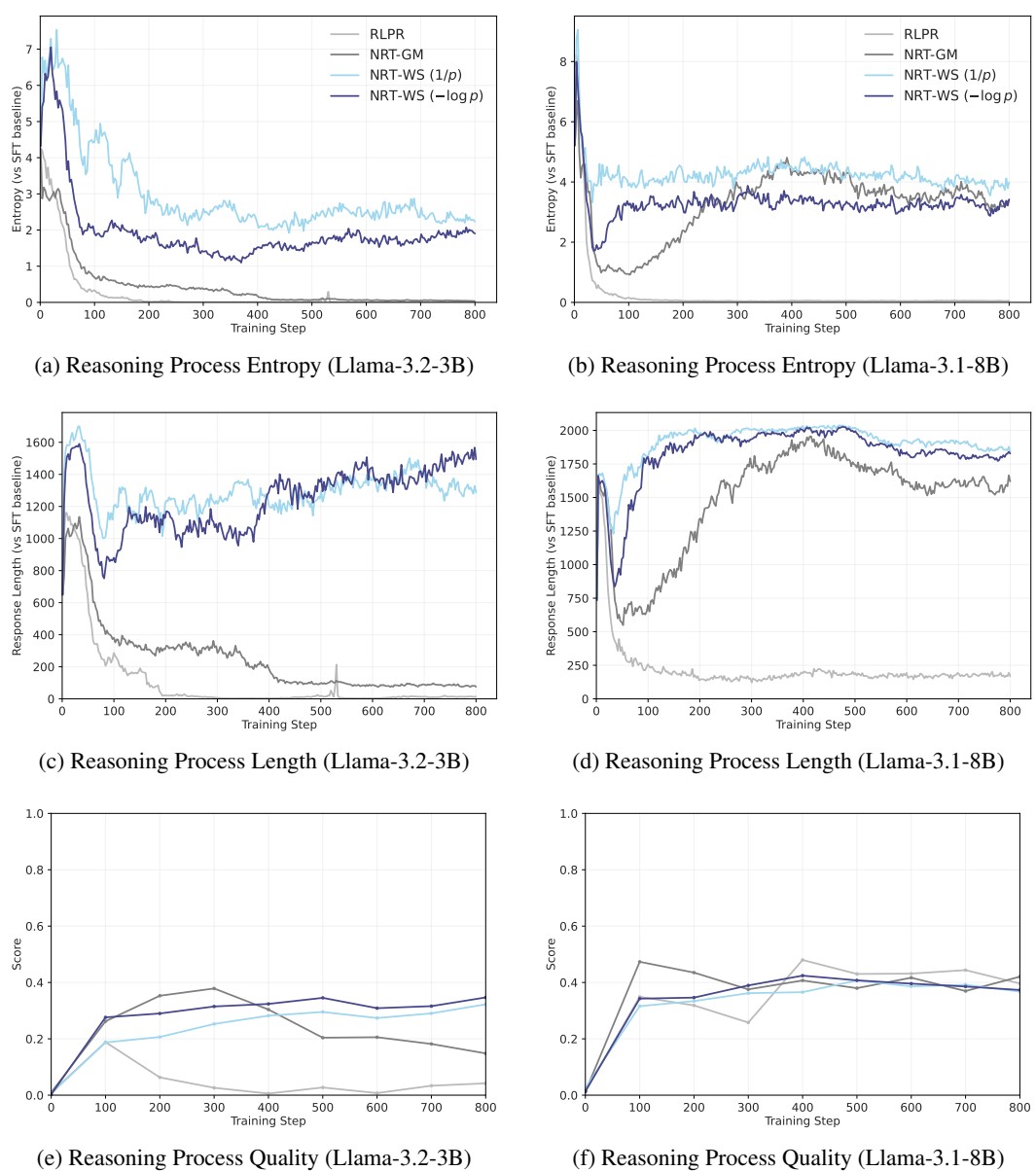

Figure 2: Evolution of the reasoning process during RL fine-tuning across three dimensions: diversity (entropy; a, b), length (c, d), and semantic quality (e, f). Quality is measured by an LLM-as-a-judge using a 0-1 score (see Appendix H). While the RLPR baseline suffers a rapid collapse across all metrics, producing short, repetitive, and low-quality reasoning, NRT variants sustain high-entropy, lengthy, and semantic reasoning on both Llama-3.2-3B and Llama-3.1-8B models. This demonstrates our approach prevents mode collapse while maintaining reasoning integrity.

ing signal where it is most needed, NRT-WS creates a powerful self-reinforcement mechanism that translates directly to the enhanced reasoning capabilities in our main results.

## 5 CONCLUSION

We introduce Native Reasoning Training (NRT), a framework that overcomes the dependencies of the SFT+RLVR paradigm. By treating reasoning as a latent variable, NRT intrinsically rewards the model for generating reasoning traces that increase its predictive confidence in the reference answer. This self-reinforcing mechanism cultivates complex reasoning using only question-answer

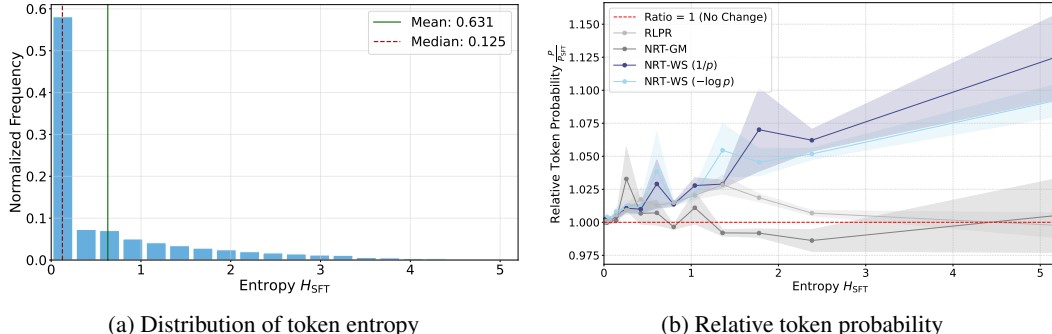

(a) Distribution of token entropy

(b) Relative token probability

Figure 3: Analysis of how NRT improves prediction of ground-truth tokens, particularly for those the SFT baseline finds most uncertain (high entropy, $H_{\text{SFT}}$). **(a)** The distribution of token entropy, showing that while most tokens are easy to predict, a long tail of high-entropy tokens exists. **(b)** Change in relative token probability ($P/P_{\text{SFT}}$). Weighted-sum NRT schemes (WS) provide the largest confidence gains for high-entropy tokens, effectively targeting the model's weaknesses.

pairs, eliminating the need for expert-written demonstrations and external verifiers. Our experiments show NRT achieves state-of-the-art results for verifier-free learning, significantly outperforming SFT baselines and prior methods in complex reasoning domains. We show that our unified framework, especially via robust reward shaping like NRT-WS to target model uncertainty, effectively resists policy collapse. Ultimately, NRT extends reinforcement learning to general and unverifiable domains, offering a scalable path toward more powerful and broadly applicable reasoning systems.

**Limitations and Future Directions.** NRT opens several exciting avenues for future exploration. The design of the reward aggregation function is a key new lever for influencing model behavior. While our work demonstrates the power of principled, handcrafted functions, we see significant potential in exploring adaptive or even fully learned reward schemes, which could further automate and enhance the reasoning acquisition process. In terms of computational resources, NRT's sampling-based approach represents a trade-off that prioritizes performance and applicability in verifier-free domains. We view this as a worthwhile investment and anticipate that future work on more sample-efficient optimization algorithms could improve its efficiency. Finally, our current study focuses on fine-tuning, but applying NRT's principles during pre-training is a compelling next step. This could instill foundational reasoning abilities directly into the model from its inception, paving the way for a new generation of more inherently capable reasoning systems.

## REPRODUCIBILITY STATEMENT

To ensure full reproducibility, we provide comprehensive details of our methodology. The theoretical framework for Native Reasoning Training (NRT), including our objective function and reward schemes, is presented in Section 3, with complete mathematical derivations in Appendices B and C. Our experimental setup is described in Section 4.1, supported by details on the dataset (Appendix D), a complete list of training hyperparameters (Appendix F), and the full training procedure in Algorithm 1 (Appendix E). All evaluations were conducted using the standardized OLMES framework, with benchmark configurations detailed in Appendix G.

### ACKNOWLEDGMENTS

This work was supported by Shanghai Artificial Intelligence Laboratory.

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

## A USE OF LARGE LANGUAGE MODELS (LLMS)

We declare the use of Large Language Models (LLMs) in this research work. The LLMs serve a supportive role in the following aspects of this project:

**Writing and Language Polishing:** LLMs assist in improving the clarity, readability, and grammatical correctness of the manuscript. This includes refining sentence structure, improving word choice, and ensuring consistent terminology throughout the paper.

**Code Development Assistance:** LLMs provide assistance in writing and debugging experimental code, including data preprocessing scripts, training pipelines, and evaluation frameworks. The models help with syntax checking, code optimization suggestions, and implementation guidance for standard machine learning practices.

**Literature Review Support:** LLMs assist in reading and summarizing research literature to identify relevant prior work and contextualize our contributions within the existing body of knowledge. This includes assistance with understanding complex technical concepts and identifying key papers in the field.

The core research ideas, experimental design, theoretical framework, and scientific contributions presented in this work are original contributions by the authors. The LLMs do not contribute to the fundamental research conception, hypothesis formulation, or interpretation of results. All experimental work, data analysis, and conclusions are conducted and drawn by the human authors.

## B FULL GRADIENT DERIVATION FOR THE NRT OBJECTIVE

This section provides a complete derivation for the policy gradient of our generalized token-level objective. The objective is designed to train a model to generate latent reasoning traces $z$ that improve its prediction of a ground-truth answer $y^\star$.

### B.1 FORMULATION OF THE TOKEN-LEVEL OBJECTIVE

Let $y^\star = (y_1^\star, \ldots, y_T^\star)$ be the ground-truth answer for a given input $x$. Our model first generates a latent reasoning trace $z$ from a policy $\pi_\theta(z|x)$. Conditioned on this trace, the model predicts the answer. For a given trace $z$, let $c_i(z, \theta)$ be the conditional probability of the $i$-th ground-truth token:

$$c_i(z, \theta) = \pi_\theta(y_i^\star|x, z, y_{<i}^\star). \tag{12}$$

We define a per-trace reward function, $R(z, \theta)$, by applying a differentiable aggregation function $f : \mathbb{R}^T \to \mathbb{R}$ to the vector of these conditional probabilities:

$$R(z, \theta) = f\big(c_1(z, \theta), c_2(z, \theta), \ldots, c_T(z, \theta)\big). \tag{13}$$

This function $f$ evaluates the quality of a trace $z$ based on how well it facilitates the prediction of the entire sequence $y^\star$. Examples for $f$ include the arithmetic mean, $f(\mathbf{c}) = \frac{1}{T} \sum_i c_i$, or the geometric mean, $f(\mathbf{c}) = (\prod_i c_i)^{1/T}$.

The overall training objective $J(\theta)$ is the on-policy expectation of this per-trace reward:

$$J(\theta) = \mathbb{E}_{z \sim \pi_\theta(z|x)}\big[R(z, \theta)\big] = \sum_z \pi_\theta(z|x)R(z, \theta). \tag{14}$$

### B.2 GRADIENT DERIVATION WITH IMPORTANCE SAMPLING

To improve sample efficiency, we employ importance sampling to enable off-policy updates using traces sampled from an older, fixed policy $\pi_{\text{old}}$. The objective in equation 14 is rewritten as:

$$J(\theta) = \mathbb{E}_{z \sim \pi_{\text{old}}(z|x)} \left[ \frac{\pi_\theta(z|x)}{\pi_{\text{old}}(z|x)} R(z, \theta) \right]. \tag{15}$$

Let $r(z, \theta) = \frac{\pi_\theta(z|x)}{\pi_{\text{old}}(z|x)}$ denote the importance ratio. Since the expectation is over a distribution independent of $\theta$, we can move the gradient operator inside:

$$\nabla_\theta J(\theta) = \mathbb{E}_{z \sim \pi_{\text{old}}(z|x)} \left[ \nabla_\theta \big(r(z, \theta)R(z, \theta)\big) \right]. \tag{16}$$

Applying the product rule for differentiation yields two terms:

$$\nabla_\theta\big(r(z,\theta)R(z,\theta)\big) = \big(\nabla_\theta r(z,\theta)\big)R(z,\theta) + r(z,\theta)\big(\nabla_\theta R(z,\theta)\big). \tag{17}$$

**Term 1: Policy Gradient Term.** The first term of equation 17 is the standard policy gradient component. By applying the log-derivative trick, $\nabla g = g\nabla\log g$, to the importance ratio, we have:

$$\nabla_\theta r(z,\theta) = r(z,\theta)\nabla_\theta\log r(z,\theta) = r(z,\theta)\nabla_\theta\log\pi_\theta(z|x). \tag{18}$$

This gives the first term of the gradient as:

$$\big(\nabla_\theta r(z,\theta)\big)R(z,\theta) = r(z,\theta)R(z,\theta)\nabla_\theta\log\pi_\theta(z|x). \tag{19}$$

This term adjusts the trace generation policy $\pi_\theta(z|x)$ to favor traces with high reward $R(z,\theta)$.

**Term 2: Differentiable Reward Term.** The second term of equation 17 accounts for the dependency of the reward function $R(z,\theta)$ on the model parameters $\theta$ through the token probabilities $c_i(z,\theta)$. Using the multivariate chain rule and the log-derivative trick, we differentiate $R(z,\theta) = f(c_1,\ldots,c_T)$:

$$\nabla_\theta R(z,\theta) = \sum_{i=1}^{T}\frac{\partial f}{\partial c_i}\cdot\nabla_\theta c_i(z,\theta) \tag{20}$$

$$= \sum_{i=1}^{T}\frac{\partial f}{\partial c_i}\cdot\big(c_i(z,\theta)\nabla_\theta\log c_i(z,\theta)\big)$$

$$= \sum_{i=1}^{T}\frac{\partial f}{\partial c_i}\,c_i(z,\theta)\nabla_\theta\log\pi_\theta(y_i^\star|x,z,y_{<i}^\star).$$

The full second term is therefore:

$$r(z,\theta)\big(\nabla_\theta R(z,\theta)\big) = r(z,\theta)\sum_{i=1}^{T}\frac{\partial f}{\partial c_i}\,c_i(z,\theta)\nabla_\theta\log\pi_\theta(y_i^\star|x,z,y_{<i}^\star). \tag{21}$$

This term constitutes a weighted supervised learning signal, where each ground-truth token's log-likelihood gradient is scaled by its contribution to the overall trace reward.

### B.3 FULL GRADIENT ESTIMATOR

Combining the expressions in equation 19 and equation 21 yields the full gradient of the objective:

$$\nabla_\theta J(\theta) = \mathbb{E}_{z\sim\pi_{\text{old}}(z|x)}\bigg[r(z,\theta)\bigg(R(z,\theta)\nabla_\theta\log\pi_\theta(z|x)$$
$$+ \sum_{i=1}^{T}\frac{\partial f}{\partial c_i}\,c_i(z,\theta)\nabla_\theta\log\pi_\theta(y_i^\star|x,z,y_{<i}^\star)\bigg)\bigg]. \tag{22}$$

In practice, we estimate this expectation using a Monte Carlo approximation with a batch of $K$ traces $\{z_k\}_{k=1}^{K}$ sampled from $\pi_{\text{old}}(z|x)$. Let $r_k(\theta) = r(z_k,\theta)$, $c_{i,k}(\theta) = c_i(z_k,\theta)$, and let $\alpha_{i,k}(\theta) = \frac{\partial f}{\partial c_i}$ be the partial derivative evaluated at $(c_{1,k}(\theta),\ldots,c_{T,k}(\theta))$. The gradient estimator is:

$$\nabla_\theta J_{\text{NRT}}(\theta) \approx \frac{1}{K}\sum_{k=1}^{K}r_k(\theta)\bigg(\underbrace{R(z_k,\theta)}_{\text{Trace Reward}}\cdot\underbrace{\nabla_\theta\log\pi_\theta(z_k|x)}_{\text{Trace Policy Gradient}}$$
$$+ \sum_{i=1}^{T}\underbrace{\alpha_{i,k}(\theta)c_{i,k}(\theta)}_{\text{Token Reward Signal}}\cdot\underbrace{\nabla_\theta\log\pi_\theta(y_i^\star|x,y_{<i}^\star,z_k)}_{\text{Token Prediction Gradient}}\bigg). \tag{23}$$

This estimator provides a principled method for training a model to generate helpful reasoning traces, as it rewards both the trace generation and token prediction policies based on their contribution to the aggregated objective. All quantities are readily computable for each trace $z_k$ in a batch.

## C GRADIENT DERIVATIONS FOR COMMON AGGREGATION FUNCTIONS

In this section, we instantiate the general gradient estimator from equation 22 for several concrete choices of the aggregation function $f$. For each case, we first define the function, then compute its partial derivative $\frac{\partial f}{\partial c_i}$ which is required for the Differentiable Reward Term. This allows us to derive a specialized form of the full gradient. The key term we derive for each case is the "Token Reward Signal", $\alpha_i c_i = \frac{\partial f}{\partial c_i} c_i(z, \theta)$.

### C.1 SEQUENCE-LEVEL LOG-PROBABILITY

Let $f$ be the sum of the log-probabilities, equivalent to the log-probability of the sequence $y^\star$.

$$R(z, \theta) = f(\mathbf{c}) = \sum_{j=1}^{T} \log c_j = \log \pi_\theta(y^\star|x, z). \tag{24}$$

The partial derivative with respect to $c_i$ is simple:

$$\frac{\partial f}{\partial c_i} = \frac{1}{c_i(z, \theta)}. \tag{25}$$

The token reward signal becomes a constant:

$$\frac{\partial f}{\partial c_i} c_i(z, \theta) = \frac{1}{c_i(z, \theta)} c_i(z, \theta) = 1. \tag{26}$$

The full gradient simplifies to:

$$\nabla_\theta J(\theta) = \mathbb{E}_{z \sim \pi_{\text{old}}(z|x)} \left[ r(z, \theta) \left( R(z, \theta) \nabla_\theta \log \pi_\theta(z|x) \right. \right.$$
$$\left. \left. + \sum_{i=1}^{T} \nabla_\theta \log \pi_\theta(y_i^\star|x, z, y_{<i}^\star) \right) \right]. \tag{27}$$

This gradient has a clean interpretation: it is a sum of the standard policy gradient term, where the reward is the log-probability of the answer, and a standard maximum likelihood (supervised) term for the answer tokens.

### C.2 SEQUENCE-LEVEL PROBABILITY

Let $f$ be the product of the conditional probabilities, which corresponds to the sequence-level probability $\pi_\theta(y^\star|x, z)$.

$$R(z, \theta) = f(\mathbf{c}) = \prod_{j=1}^{T} c_j = \pi_\theta(y^\star|x, z). \tag{28}$$

The partial derivative of $f$ with respect to $c_i$ is the product of all other probabilities:

$$\frac{\partial f}{\partial c_i} = \prod_{j \neq i} c_j = \frac{\prod_{j=1}^{T} c_j}{c_i} = \frac{R(z, \theta)}{c_i(z, \theta)}. \tag{29}$$

The token reward signal for the $i$-th token is therefore:

$$\frac{\partial f}{\partial c_i} c_i(z, \theta) = \frac{R(z, \theta)}{c_i(z, \theta)} c_i(z, \theta) = R(z, \theta). \tag{30}$$

Substituting this into the general gradient formula equation 22, we can factor out $R(z, \theta)$:

$$\nabla_\theta J(\theta) = \mathbb{E}_{z \sim \pi_{\text{old}}(z|x)} \left[ r(z, \theta) R(z, \theta) \left( \nabla_\theta \log \pi_\theta(z|x) \right. \right.$$
$$\left. \left. + \sum_{i=1}^{T} \nabla_\theta \log \pi_\theta(y_i^\star|x, z, y_{<i}^\star) \right) \right], \tag{31}$$

where the trace reward $R(z, \theta)$ acts as a scaling factor for both the trace policy gradient and the standard supervised gradients for the answer tokens.

## C.3 GEOMETRIC MEAN

Let $f$ be the geometric mean of the conditional probabilities.

$$R(z, \theta) = f(\mathbf{c}) = \left( \prod_{j=1}^{T} c_j \right)^{1/T}. \tag{32}$$

Using the chain rule, the partial derivative with respect to $c_i$ is:

$$\frac{\partial f}{\partial c_i} = \frac{1}{T} \left( \prod_{j=1}^{T} c_j \right)^{\frac{1}{T}-1} \cdot \left( \prod_{j \neq i} c_j \right) = \frac{1}{T} \frac{\left( \prod_{j=1}^{T} c_j \right)^{1/T}}{c_i} = \frac{R(z, \theta)}{T \cdot c_i(z, \theta)}. \tag{33}$$

The token reward signal is therefore scaled by $1/T$:

$$\frac{\partial f}{\partial c_i} c_i(z, \theta) = \frac{R(z, \theta)}{T \cdot c_i(z, \theta)} c_i(z, \theta) = \frac{R(z, \theta)}{T}. \tag{34}$$

Similar to the sequence-level probability case, we can factor out the reward $R(z, \theta)$:

$$\nabla_\theta J(\theta) = \mathbb{E}_{z \sim \pi_{\text{old}}(z|x)} \left[ r(z, \theta) R(z, \theta) \left( \nabla_\theta \log \pi_\theta(z|x) \right. \right.$$
$$\left. \left. + \frac{1}{T} \sum_{i=1}^{T} \nabla_\theta \log \pi_\theta(y_i^\star | x, z, y_{<i}^\star) \right) \right]. \tag{35}$$

The gradient has a similar form to the sequence-level probability case, but the supervised signal is averaged.

## C.4 ARITHMETIC MEAN

Let $f$ be the arithmetic mean of the conditional probabilities.

$$R(z, \theta) = f(\mathbf{c}) = \frac{1}{T} \sum_{j=1}^{T} c_j. \tag{36}$$

The partial derivative with respect to $c_i$ is a constant:

$$\frac{\partial f}{\partial c_i} = \frac{1}{T}. \tag{37}$$

The token reward signal for the $i$-th token is:

$$\frac{\partial f}{\partial c_i} c_i(z, \theta) = \frac{c_i(z, \theta)}{T}. \tag{38}$$

Plugging this into the general formula gives the full gradient:

$$\nabla_\theta J(\theta) = \mathbb{E}_{z \sim \pi_{\text{old}}(z|x)} \left[ r(z, \theta) \left( R(z, \theta) \nabla_\theta \log \pi_\theta(z|x) \right. \right.$$
$$\left. \left. + \frac{1}{T} \sum_{i=1}^{T} c_i(z, \theta) \nabla_\theta \log \pi_\theta(y_i^\star | x, z, y_{<i}^\star) \right) \right]. \tag{39}$$

In this case, the supervised gradient for each token $y_i^\star$ is weighted by its own conditional probability $c_i(z, \theta)$, encouraging the model to focus on tokens it is already confident about.

## C.5 WEIGHTED SUM

Let $f$ be a weighted sum of the conditional probabilities, where $\{w_i\}_{i=1}^T$ are non-negative weights.

$$R(z, \theta) = f(\mathbf{c}) = \sum_{j=1}^{T} w_j c_j. \tag{40}$$

The partial derivative with respect to $c_i$ is simply the corresponding weight:

$$\frac{\partial f}{\partial c_i} = w_i. \tag{41}$$

The token reward signal for the $i$-th token is:

$$\frac{\partial f}{\partial c_i} c_i(z, \theta) = w_i c_i(z, \theta). \tag{42}$$

The full gradient is a generalization of the arithmetic mean case:

$$\nabla_\theta J(\theta) = \mathbb{E}_{z \sim \pi_{\text{old}}(z|x)} \Bigg[ r(z, \theta) \bigg( R(z, \theta) \nabla_\theta \log \pi_\theta(z|x) \\ + \sum_{i=1}^{T} w_i c_i(z, \theta) \nabla_\theta \log \pi_\theta(y_i^\star | x, z, y_{<i}^\star) \bigg) \Bigg]. \tag{43}$$

This formulation allows for explicit control over the importance of each token in the supervised learning objective, for example, by assigning higher weights $w_i$ to more critical tokens in the answer. The arithmetic mean is a special case where $w_i = 1/T$ for all $i$.

## D DATASET DETAILS

The primary training dataset used across all our experiments is a 200K-sample subset randomly selected from `tulu-3-sft-mixture` (Lambert et al., 2024). This is a diverse, large-scale collection of high-quality instruction-following data. A key characteristic of this dataset is that it exclusively contains question-answer pairs $(x, y^\star)$ and does not include any human-annotated "gold" reasoning traces ($z^\star$). The detailed composition of our 200K-sample subset is provided in Table 4.

**Comparison with Baseline Dataset.** A crucial distinction between our NRT approach and the baseline methods lies in the training data and the nature of the responses used during optimization. Table 5 provides a summary of these differences. While all methods operate in a verifier-free setting (i.e., without requiring external tools to check reasoning steps), their underlying data assumptions and response characteristics vary significantly:

**JLB** (Tang et al., 2025) is demonstrated on "unverifiable" data, specifically `Numina-proof`, which contains long-form proofs without a distinct, short-form answer for automated checking. This differs from our setup, where the ground truth is a short, verifiable answer $y^\star$.

**Verifree** (Zhou et al., 2025) utilizes `WebData`, a custom 61K sample dataset derived from `WebInstruct`. This dataset is intentionally curated to have very short answers (fewer than 7 tokens), enforcing a strong constraint on the response length.

**RLPR** (Yu et al., 2025b) is trained on a 77K non-mathematical prompt set. While it generates some reasoning, the average response length remains short (around 13 tokens).

**NRT (Ours)**, in contrast, is trained on a general-purpose instruction-following dataset (`tulu-3-sft-mixture`) that lacks pre-existing reasoning steps. Unlike methods designed for short-form outputs, our training data's reference answers ($y^\star$) are often extensive, averaging over 400 tokens. The model must therefore learn to generate its own reasoning traces "natively" to bridge the gap from prompt to these complex answers. This highlights NRT's ability to elicit reasoning from standard question-answer pairs, a key departure from baselines that either rely on specialized long-form data (JLB) or are constrained to short responses (Verifree, RLPR).

Table 4: Composition of the 200k-sample training dataset, randomly selected from the `tulu-3-sft-mixture`.

| Source | Count | Percentage |
|---|---|---|
| ai2-adapt-dev/personahub_math_v5_regen_149960 | 33,036 | 16.13% |
| ai2-adapt-dev/evol_codealpaca_heval_decontaminated | 23,778 | 11.61% |
| ai2-adapt-dev/tulu_v3.9_aya_100k | 22,088 | 10.79% |
| ai2-adapt-dev/flan_v2_converted | 20,010 | 9.77% |
| ai2-adapt-dev/tulu_v3.9_wildchat_100k | 19,378 | 9.46% |
| ai2-adapt-dev/numinamath_tir_math_decontaminated | 13,999 | 6.84% |
| allenai/tulu-3-sft-personas-math-grade | 11,115 | 5.43% |
| ai2-adapt-dev/tulu_v3.9_open_math_2_gsm8k_50k | 11,081 | 5.41% |
| ai2-adapt-dev/tulu_v3.9_wildjailbreak_decontaminated_50k | 11,069 | 5.40% |
| ai2-adapt-dev/tulu_v3.9_synthetic_finalresp_wildguardmixtrain... | 10,935 | 5.34% |
| ai2-adapt-dev/personahub_code_v2_34999 | 7859 | 3.84% |
| ai2-adapt-dev/personahub_ifdata_manual_seed_v3_29980 | 6483 | 3.17% |
| ai2-adapt-dev/tulu_v3.9_personahub_math_interm_algebra_20k | 4529 | 2.21% |
| ai2-adapt-dev/coconot_converted | 2441 | 1.19% |
| ai2-adapt-dev/tulu_v3.9_sciriff_10k | 2177 | 1.06% |
| ai2-adapt-dev/no_robots_converted | 2114 | 1.03% |
| ai2-adapt-dev/oasst1_converted | 1606 | 0.78% |
| ai2-adapt-dev/tulu_v3.9_table_gpt_5k | 1062 | 0.52% |
| ai2-adapt-dev/tulu_hard_coded_repeated_10 | 40 | 0.02% |
| **TOTAL** | **204800** | **100.00%** |

Table 5: Comparison of verifier-free RL methods. This table details each method's aggregation function $f(\mathbf{c})$, reward ormalization, and the characteristics of the responses used for training. Note the significant difference in average response length between our NRT methods, which generate long reasoning traces spontaneously, and the baselines, which often use datasets with short or specialized responses.

| Method | Aggregation Function | Reward Normalization | Average Response Length |
|---|---|---|---|
| **JLB** | $f(\mathbf{c}) = \sum_{j=1}^{T} \log c_j$ | RLOO | Unknown. |
| **Verifree** | $f(\mathbf{c}) = \prod_{j=1}^{T} c_j$ | RLOO | $\leq 7$ tokens. |
| **RLPR** | $f(\mathbf{c}) = \frac{1}{T} \sum_{j=1}^{T} c_j$ | Uses a clipped reward $r_{final} = \text{clip}(r - r', 0, 1)$, where $r'$ is the score for generating $y^\star$ without reasoning $z$. | 12.52 tokens. |
| **NRT-GM** | $f(\mathbf{c}) = \left(\prod_{j=1}^{T} c_j\right)^{1/T}$ | Normalized advantage of the clipped reward difference, $\max(0, r - r')$. where $r'$ is the score for generating $y^\star$ with empty reasoning $\emptyset$. | 415.97 tokens. |
| **NRT-WS** | $f(\mathbf{c}) = \sum_{j=1}^{T} w_j c_j$ | Normalized advantage of the clipped reward difference, $\max(0, r - r')$. where $r'$ is the score for generating $y^\star$ with empty reasoning $\emptyset$. | 415.97 tokens. |

---

**Algorithm 1** Native Reasoning Training (NRT)

---

**Input:**
    Pre-trained language model $\pi_\theta$.
    Training dataset $\mathcal{D} = \{(x, y^\star)\}$.
    Differentiable aggregation function $f$ (see Table 1).
    Hyperparameters: batch size $B$, traces per prompt $K$, learning rate $\eta$, format loss weight $\lambda_{\text{format}}$.
**Output:**
    Trained reasoning model $\pi_\theta$.
**Initialize:**
1: Initialize sampling model: $\pi_{\text{old}} \leftarrow \pi_\theta$.
2: **for** each training step **do**
3:     Sample a minibatch $\mathcal{B} = \{(x_j, y_j^\star)\}_{j=1}^{B}$ from $\mathcal{D}$.
4:     Sample a group of $K$ traces $\{z_k\}_{k=1}^{K} \sim \pi_{\text{old}}(z|x, t_{\text{start}})$.
5:     Append $t_{\text{end}}$ to any incomplete trace $z_k$.
6:     Calculate baseline reward: $R_{\text{base}} \leftarrow f(\{\pi_{\text{old}}(y_i^\star | x, \emptyset, y_{<i}^\star)\}_{i=1}^{T})$.
7:     Calculate trace reward: $R_k \leftarrow f(\{\pi_\theta(y_i^\star | x, z_k, y_{<i}^\star)\}_{i=1}^{T})$.
8:     Compute clipped reward: $R_k' \leftarrow \max(0, R_k - R_{\text{base}})$ (Eq. 9).
9:     Compute advantages: $\{A_k\}_{k=1}^{K} \leftarrow \frac{R' - \text{mean}(R')}{\text{std}(R')}$ (Eq. 10).
10:                                                ▷ Compute NRT loss gradient
11:     Calculate token reward signals: $\{S_{i,k}\}_{i=1}^{T} \leftarrow \{\frac{\partial f}{\partial c_i} c_{i,k}\}_{i=1}^{T}$.
12:     $\nabla L_{k,\text{NRT}} \leftarrow -A_k \nabla_\theta \log \pi_\theta(z_k|x) - \sum_{i=1}^{T} S_{i,k} \nabla_\theta \log \pi_\theta(y_i^\star | x, y_{<i}^\star, z_k)$.
13:                                             ▷ Compute format loss gradient
14:     $\nabla L_{k,\text{format}} \leftarrow -\nabla_\theta\big(\log \pi_\theta(t_{\text{start}}|x) + \log \pi_\theta(t_{\text{end}}|x, z_k)\big)$.
15:     $\nabla L_{\text{total}} \leftarrow \sum_k [\nabla L_{k,\text{NRT}} + \lambda_{\text{format}} \nabla L_{k,\text{format}}]$.
16:     Update model parameters: $\theta \leftarrow \theta - \eta \cdot \nabla L_{\text{total}}$.
17:     Update sampling model: $\pi_{\text{old}} \leftarrow \pi_\theta$.
18: **end for**
19: **return** $\pi_\theta$.

---

# E    THE NRT TRAINING ALGORITHM

The Native Reasoning Training process is implemented as an off-policy reinforcement learning loop, detailed in Algorithm 1. The training cycle begins by initializing the policy $\pi_\theta$ from a suitable SFT checkpoint. Then, for each prompt $(x, y^\star)$ in a training batch, the following steps are executed.

First, a group of $K$ candidate reasoning traces $\{z_k\}$ is sampled from a frozen, older policy $\pi_{\text{old}}$. This decouples data generation from the gradient update, enhancing stability. Next, an intrinsic reward is calculated for each trace by evaluating the current policy's confidence in predicting the reference answer $y^\star$. To reduce variance, these rewards are stabilized using the GRPO-inspired advantage mechanism from Section 3.2.2: rewards are clipped against a no-reasoning baseline and then normalized group-wise to produce a final advantage estimate $A_k$.

Finally, the advantage is used to compute a composite gradient signal as defined in Equation 8. This gradient combines a policy gradient update for the entire trace $z_k$ with a weighted supervised update for the answer tokens $y^\star$. A supplementary format-supervision loss helps enforce the desired output structure. After the model parameters $\theta$ are updated, the sampling policy $\pi_{\text{old}}$ is synchronized with the new policy, completing the loop. This iterative process allows the model to progressively refine its own reasoning abilities without external trace supervision.

# F    TRAINING HYPERPARAMETERS AND IMPLEMENTATION DETAILS

All our Reinforcement Learning experiments, including the NRT variants and re-implemented RL baselines, are trained using the GRPO algorithm (Shao et al., 2024) based on the VERL (Sheng et al., 2024) training framework. To ensure a fair comparison, the re-implemented RL baselines incorporate the same reward stabilization (Section 3.2.2) and structural format supervision (Section 3.2.3) as our NRT methods. This was a necessary adaptation, as the original RL methods were designed for

instruction-tuned models and are not directly applicable to the base models used in our experiments, which require explicit guidance to generate structured reasoning. We leverage Dr.GRPO (Liu et al., 2025) for length normalization, which is designed to enhance stability and performance. The models are trained for a single epoch over the 200K-sample dataset.

Table 6: Hyperparameters for NRT and baseline RL training.

| Parameter | Value |
|---|---|
| *General Training Configuration* | |
| Learning Rate | 1e-5 |
| Learning Rate Scheduler | constant |
| Weight Decay | 0.01 |
| Training Epochs | 1 |
| Training Batch Size | 256 |
| Max Prompt Length | 4096 |
| Max Response Length | 4096 |
| Thinking Format SFT Loss Coef. | 0.3 |
| *GRPO Algorithm Parameters* | |
| Advantage Estimator | GRPO |
| Gamma | 1 |
| Length Normalization | Dr.GRPO |
| PPO Mini-batch Size | 64 |
| PPO Clip Ratio (Low) | 0.2 |
| PPO Clip Ratio (High) | 0.28 |
| *Generation* | |
| Rollout Backend | vLLM |
| Generations per Prompt ($N$) | 8 |
| Temperature | 1 |
| Top_P | 1 |
| Max Generation Tokens (Reasoning) | 2048 |
| *Regularization* | |
| KL Loss Coefficient | 0.0 |
| Entropy Coefficient | 0.0 |

During the RL rollout phase, we generate 8 responses for each prompt in the training batch, with each generated reasoning trace capped at a maximum of 2048 tokens. This process is accelerated using the vLLM inference engine (Kwon et al., 2023). To encourage the model to explicitly generate its reasoning process within the desired structure, we incorporate an auxiliary SFT loss with a coefficient of 0.3. This loss incentivizes the model to use the `<|think_start|>` and `<|think_end|>` special tokens. We did not employ KL divergence or entropy regularization in our experiments. The key hyperparameters for our training setup are summarized in Table 6.

## G  EVALUATION BENCHMARKS DETAILS

To ensure our results are rigorous, reproducible, and comparable to prior and future work, we adopt the Open Language Model Evaluation Standard (OLMES) toolkit for all benchmark evaluations [1]. OLMES is a standardized, open-source framework designed to mitigate common sources of variance in LLM evaluation, such as subtle differences in prompt formatting, the choice of few-shot examples, and score normalization techniques (Gu et al., 2024). By using a unified evaluation harness, we can confidently attribute performance differences to our training methods rather than to inconsistencies in the evaluation setup. Our evaluation suite comprises nine diverse benchmarks, selected to provide a holistic assessment of model capabilities across general knowledge, reasoning, truthfulness, mathematics, reading comprehension, code generation, and instruction following. The specific configuration for each benchmark, including the prompting strategy, number of in-context examples

---

[1]`https://github.com/allenai/olmes`

Table 7: Details of the evaluation benchmarks and their configurations within the OLMES framework. CoT denotes Chain-of-Thought prompting.

| Benchmark | Domain | Prompting Strategy | Shot Count | Metric |
|---|---|---|---|---|
| MMLU | General Knowledge | Zero-shot CoT | 0 | Accuracy |
| BBH | General Reasoning | CoT | 3 | Accuracy |
| DROP | Reading Comprehension | Zero-shot Chat | 0 | F1 Score |
| PopQA | Fact-based Q&A | Standard Prompting | 15 | Accuracy |
| TruthfulQA | Truthfulness | Standard Prompting | 6 | Accuracy |
| GSM8K | Mathematical Reasoning | Zero-shot CoT | 0 | Accuracy |
| MATH | Mathematical Reasoning | Zero-shot Reasoning | 0 | Accuracy |
| HumanEval | Code Generation | Standard Prompting | 0 | Pass@10 |
| IFEval | Instruction Following | Standard Prompting | 0 | Pass@10 |

(shots), and primary evaluation metric, is detailed in Table 7. All configurations are standard settings provided within the OLMES framework.

## H  Semantic Quality Evaluation Details

To assess the intrinsic quality of the reasoning processes generated during training, we implement an automated evaluation pipeline using the **grok-4.1-fast** model as a judge. We prioritize identifying logical coherence and faithfulness in the reasoning trace rather than strictly checking the final answer format. To ensure reproducibility and minimize variance in the scoring, we set the temperature to 0.0. For every model checkpoint analyzed, we randomly sample $N = 100$ instances from the validation set and query the judge model. The specific prompt template used for this evaluation is detailed in Table 8; inputs are inserted into the placeholders {problem}, {reasoning}, and {ground_truth} prior to inference.

## I  Qualitative Analysis of Native Reasoning Models

This section provides a more in-depth qualitative analysis of the reasoning processes learned by NRT models, supplementing the quantitative results in the main paper. We first perform an aggregate analysis of the vocabulary that emerges in the generated reasoning traces. We then present specific case studies that illustrate how this learned reasoning corrects the failures of baseline models on individual problems, demonstrating the practical impact of our approach.

### I.1  Emergent Vocabulary of Reasoning

To better understand what the model learns to "think," we analyze the vocabulary of the reasoning traces ($z$) generated by our best-performing model, NRT-WS ($-\log p$), and compare it to the vocabulary of the ground-truth answers ($y^\star$).

NRT autonomously develops a distinct "language of reasoning" rich with procedural and meta-cognitive terms. As shown in the word cloud in Figure 4b, the generated reasoning processes are dominated by words associated with problem-solving procedures, such as "let", "step", "solve", "given", and "using". This indicates the model is not merely repeating the problem statement but is actively constructing a plan of attack. More revealingly, the frequency analysis in Figure 4a highlights the words whose usage increases most dramatically in the reasoning trace compared to the ground-truth data. Terms like "premise," "developed," "reasoning," and "mentioned" show the highest increase, suggesting the model has learned to externalize its own meta-cognitive process: laying out premises, explaining its logic, and framing its internal queries.

The model successfully learns to disentangle the reasoning process from the final answer, confining answer-specific formatting to the output. While the reasoning and ground-truth word clouds (Figures 4b and 4d) share core problem-solving terms, there is a clear separation of function. The

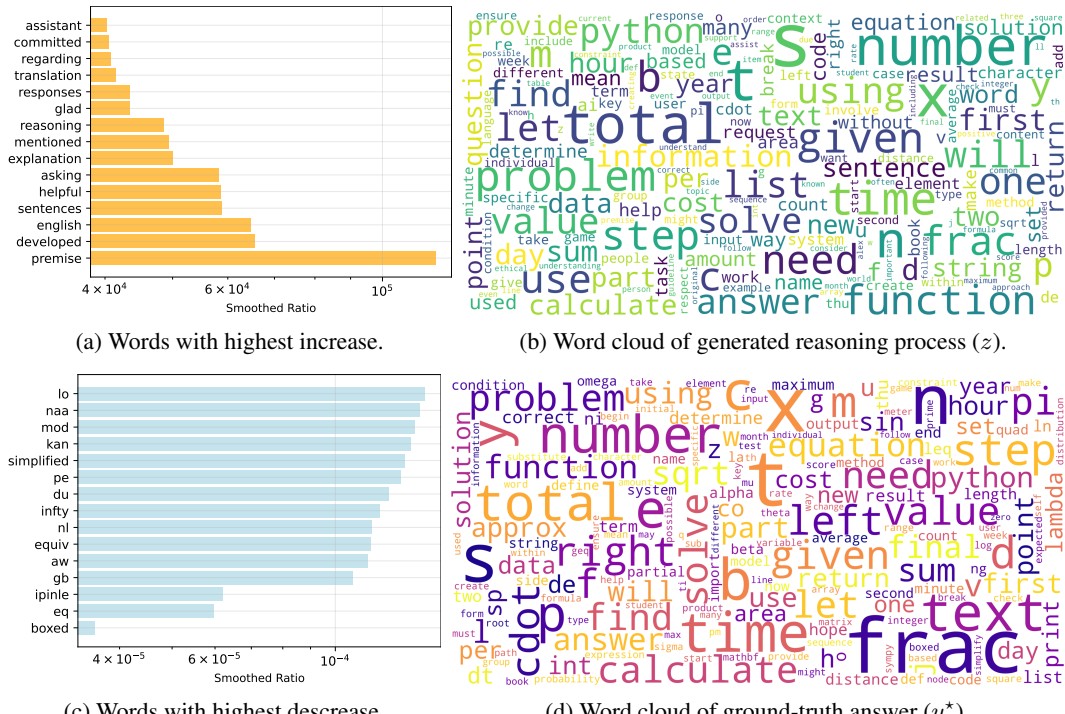

(a) Words with highest increase.

(b) Word cloud of generated reasoning process ($z$).

(c) Words with highest descrease.

(d) Word cloud of ground-truth answer ($y^\star$).

Figure 4: Qualitative analysis of the vocabulary learned by NRT-WS ($-\log p$). **(b)** The word cloud of the generated reasoning process is rich with procedural terms like 'let', 'step', and 'solve'. **(a)** NRT specifically learns to use meta-cognitive words like 'premise' and 'explanation' in its reasoning. **(d)** The ground-truth word cloud is more focused on the problem's nouns and final answer. **(c)** Conversely, NRT learns to suppress answer-specific formatting like 'boxed' from its reasoning traces. Together, these show NRT organically develops a distinct "language of reasoning".

frequency analysis in Figure 4c shows that tokens overwhelmingly associated with final answers, such as the formatting token "boxed" (used in math problems to demarcate the final numerical answer), are effectively suppressed in the reasoning trace. This demonstrates that NRT, without any direct supervision on vocabulary, successfully teaches the model to differentiate between the "thinking" part of the task and the "answering" part, correctly adhering to the intended latent reasoning structure. This learned separation is a key factor in its ability to generate clean, step-by-step logic that leads to improved final answers.

## I.2 CASE ANALYSIS OF GROUNDTRUTH TOKEN PROBABILITIES

Our hypothesis is that a successful training method should not only lead to the correct answer but also increase the model's confidence in that answer by generating a sound reasoning trace. The NRT objective, $\mathbb{E}_{z \sim \pi_\theta(\cdot|x)}[\log \pi_\theta(y^\star|x, z)]$, is designed expressly for this purpose: it rewards the generation of reasoning processes ($z$) that make the ground-truth answer ($y^\star$) more probable.

To provide a concrete, quantitative example of this mechanism, we analyze a single prediction from a fact-based question-answering task. Table 9 compares the performance of the baseline SFT model against our NRT-trained models and other baselines. While most models produce the correct year, the models trained to reason first generate an internal monologue that weighs evidence and historical facts.

The result is a striking increase in predictive confidence for the best model. The standard SFT model is uncertain, assigning only a 30.8% probability to the correct answer, "1852". In contrast, after generating its native reasoning trace by citing external knowledge and evidence, our NRT-WS ($1/p$) model's confidence in the same answer jumps to 50.4%, a relative increase of over 63%. This case study provides direct evidence that NRT can successfully train the model to generate reasoning

processes that strengthen its own conviction in the correct solution, validating the core principle of our approach.

### I.3 CASE STUDY OF GENERATED RESPONSES

To provide a qualitative illustration of how different training methods affect the model's emergent reasoning, we present a case study in Table 10. The example features an open-ended, ambiguous query that does not have a single verifiable "correct" answer, making it an excellent test bed for practical reasoning ability.

The SFT model responds directly without a reasoning process, acknowledging the question's ambiguity and offering to help if more information is provided. While polite, it is not proactive.

The RLPR model, trained with a basic intrinsic reward, develops a metacognitive reasoning process to "break down the key factors to consider." However, its reasoning remains high-level, and the resulting answer offers only generic institutional examples (e.g., Harvard Business School) rather than specific, actionable course recommendations.

In contrast, our NRT models leverage the latent reasoning process to navigate the query's vagueness more effectively. The NRT-GM model reasons about general evaluation criteria like curriculum and instructor quality, leading to a helpful final answer that outlines the structure of a hypothetical course. The NRT-WS models exhibit an even more exploratory internal monologue, actively brainstorming potential interpretations and considering relevant entities (e.g., Udemy, Coursera, ICF). For NRT-WS (1/p), this richer reasoning allows it to synthesize a far more useful final answer with specific, actionable suggestions. Interestingly, while NRT-WS (-log p) follows a similar exploratory reasoning path, its final answer confidently hallucinates a non-existent program called "the 7 Module Coaching Course." This overall comparison demonstrates that NRT fosters a more robust and practical form of reasoning, where the model learns to think through ambiguity to generate helpful and comprehensive responses.

Table 8: The complete prompt template used for the LLM-as-a-judge semantic evaluation. The prompt is fed to the model as a single continuous context.

---

**Evaluation Prompt**

---

You are verifying the *reasoning process* quality, not the final answer formatting.

Score the quality of the thought process for the given reasoning trace. Focus on:
1. Logical coherence and absence of contradictions.
2. Faithfulness to the provided problem statement.
3. Correct mathematical or algorithmic steps (even if arithmetic slips occur later).
4. Absence of hallucinated facts or unjustified leaps.

Return JSON with fields:
{
    "score": float in [0, 1],
    "justification": "one paragraph that cites concrete issues or strengths",
    "flags": ["optional short tags such as `hallucination`, `missing-step`, ...]
}

### Problem:
```

{problem}
```

### Model reasoning trace:
```

{reasoning}
```

### Ground-truth answer:
```

{ground_truth}
```

Give higher scores when the reasoning is consistent and well-grounded, and lower scores when it is self-contradictory, irrelevant, or clearly flawed.

---

Table 9: Case study on the impact of native reasoning on predictive confidence. For the same query, we compare the final answer probability from a standard SFT model and models trained with different reasoning-based objectives. Models that generate an internal reasoning process show significantly boosted confidence in the correct answer.

| Model | Generated Reasoning Process (z) | $P(y^*|x,z)$ |
|---|---|---|
| **Query** $(x)$ 

 **Reference Answer** $(y^*)$ | Questions: ['when did luois braille die?'] 
 Solution: 
 1852 | |
| SFT | `N/A` | 0.3083 |
| RLPR | `<|think_start|>` Sure, let's determine the solution for the input "Questions: ['when did luois braille die? ']" and output "March 6, **1852**" based on the reason "The output answer is the exact answer for the given questions." `<|think_end|>` | 0.0375 |
| NRT-GM | `<|think_start|>` To answer the task "New input: Questions: ['when did luois braille die?'] Solution:" ... It appears to be a fact-finding or historical question that can be answered with a date. The solution provided is needed to match the format and substance of the task, which asks for a fact known to be true. The task is an example of fact-based question-answer task in the form of a natural language question and its expected answer. According to historical account, Louis Braille, a French educator and inventor of a reading and writing system for those who are blind, died in **1852** at the age of 43. Given that the task is to answer the question "when did Louis Braille die?", the correct answer is : ... `<|think_end|>` | 0.1354 |
| NRT-WS $(1/p)$ | `<|think_start|>` This question is about a person (Luois Braille) that was born in 1809 and died in **1852** (cf Wikipedia "Louis Braille", snippet: "Louis Braille (4 January 1809 – 6 January **1852**)..."). You don't need to understand Natural Language Processing to do this, you just need to search for the answer online... Evidence: Louis Braille, the inventor of braille, died on January 6, **1852**, at the age of 43... It is clear that Louis Braille, the creator of the braille system of writing, died on January 6, **1852**... `<|think_end|>` | **0.5042** |
| NRT-WS $(-\log p)$ | `<|think_start|>` To answer "when did louis braille die?" we need to consult knowledge about Louis Braille, who invented the braille script. He died in **1852**... For Louis Braille, he was a man who made a lifesaving system in this world for those with vision problems. Louis Braille died on January 6, **1852**. So, the answer is **1852**... Louis Braille was born on 4 January 1809 in Coupvray, Île-de-France. He died on 6 January **1852** at the age of forty-two... Louis Braille, the inventor of the Braille system for the visually impaired, died on January 6, **1852**... The solution as per your need: 6 January **1852**...`<|think_end|>` | 0.3150 |

Table 10: A case study comparing the generated reasoning process ($z$) for different methods on a sample query from the training dataset.

| Query (x): What is a good 7 module coaching course? | |
| --- | --- |
| **Method** | **Generated Reasoning Process (z) and Generated Answer (y)** |
| SFT | It's great that you're looking to learn more and improve your skills through a 7-module coaching course! However, without specific criteria or your area of interest, it's challenging to recommend a particular course. There are many excellent coaching courses available on various topics, including leadership, personal development, career coaching, and more. If you could provide more details about your specific area of interest or the type of coaching you're looking for, I'd be happy to suggest some options. |
| RLPR | **Reasoning:** To find a good 7 module coaching course, let's break down the key factors to consider when selecting a coaching course with 7 modules. 

 **Answer:** Determining the "best" 7 module coaching course can depend on a variety of factors, such as the specific goals of the individual or organization... here are a few examples of 7 module coaching courses that are highly regarded in various fields... A well-regarded course in this category could be offered by the Harvard Business School Executive Education or the Leadership Institute... It's always a good idea to research the course content, the credentials of the instructors, and the reputation of the provider before making a decision. |
| NRT-GM | **Reasoning:** To answer the question "What is a good 7 module coaching course?"... I can't provide a specific recommendation... without additional context about what type of coaching you're interested in... A good coaching course should cover a comprehensive curriculum, be taught by experienced and qualified professionals, and offer practical exercises... You might want to consider the following criteria when evaluating a coaching course: Course Content... Instructor Qualifications... Reviews and Ratings... 

 **Answer:** A good 7 module coaching course could cover a variety of topics depending on the specific area of coaching you're interested in. Here's a general outline of what a 7 module coaching course might look like... Overall, a good 7 module coaching course would provide a comprehensive introduction to the field of coaching, covering the core skills, theories, and ethical considerations necessary for effective coaching practice. |
| NRT-WS ($1/p$) | **Reasoning:** To answer... it's important to understand that the "best" coaching course will depend on your specific needs, interests, and goals. There are many online platforms offering coaching courses, such as Udemy, Coursera... Some popular 7-module coaching courses include "Coaching Skills Foundations" by the International Coach Federation (ICF)... "The Life Coach School: The Ultimate Life Coach Training Program" by Brooke Castillo. These courses cover various aspects of coaching... It's also important to consider the accreditation... 

 **Answer:** There are many coaching courses available, and the "goodness" of a course can vary... However, a good 7 module coaching course should cover a range of topics including coaching techniques, ethical considerations, client communication, goal setting, and more... Some popular coaching courses include those offered by the International Coach Federation (ICF), the Institute for Professional Excellence in Coaching (iPEC), and the Co-Active Training Institute. Remember to choose a course that aligns with your career goals and interests. |
| NRT-WS ($-\log p$) | **Reasoning:** To provide a recommendation for a good 7 module coaching course, I need more context... For a general coaching course, here are some popular options: 1. The Life Coach School... 2. The Tony Robbins Coaching Certification Program... 3. The Co-Active Coaching Certification Program... A good 7 module coaching course can vary depending on the specific area... platforms like Coursera, Udemy, and LinkedIn Learning offer a wide range of coaching courses... 

 **Answer:** There are many excellent coaching courses available, but the "7 Module Coaching Course" is a comprehensive and highly recommended program. It is designed to provide individuals with the skills and knowledge needed to become effective coaches... The course is structured into seven modules, each focusing on a different aspect of coaching, including communication, goal setting, motivation, and more... |

