# OpenReview forum: "Native Reasoning Models: Training Language Models to Reason on Unverifiable Data"
_ICLR.cc/2026/Conference — ICLR 2026 Poster_

### Official Review · Reviewer_iE85 · 2025-10-26

**Soundness:** 2
**Presentation:** 3
**Contribution:** 2
**Rating:** 4
**Confidence:** 3

**Summary:**

This paper introduces a unified verifier-free framework (NRT) that treats the reasoning trace as a latent variable and maximizes an intrinsic reward computed only from question-answer (QA) pairs without expert CoT or external verifiers. The authors say this re-frames prior methods, exposes failure modes (e.g., policy collapse), and yields new reward aggregations, etc and present several experiments focusing on the llama model family.

**Strengths:**

* The paper presents a unified formulation that subsumes several verifier-free ideas into one token-level objective and gradient, making design levers more explicit by casting them as choices of $f$, which is a helpful perspective.

* NRT relies only on QA pairs which can potentially help lessen/reduce dependence on costly human chain-of-thought (CoT) annotations or judge models.

* It appears that NRT might avoids the mode collapse observed in RLPR, maintaining diverse reasoning traces while still converging on higher answer accuracy.

**Weaknesses:**

* The experimentation is quite limited in restricting attention to just two models within the same model family; it's unclear how well these results can generalize to other more model families as a result. Results on bigger models, alternative corpora, and domains with very long outputs or strong factuality constraints would strengthen the claims.

* The results in Table 3 do not seem to show consistent or state-of-the-art results by NRT as claimed in the paper; the results seem mixed as claims of significant gains are made only on the average score. The individual scores tell quite a different story. Moreover, it's unclear what variant of NRT to use consistently over others and why as the results, again, seem to be mixed.
  * Just as one example, TruthfulQA hardly moves in the results; if NRT induced genuinely more truthful reasoning, I would expect clearer TruthfulQA gains, no?

* The paper also notes prior baselines use different datasets and response lengths (Appendix D, p18), while NRT trains on long-answer QA (longer avg response). From this, it seems like the data regimes also differ across prior work; Appendix D (Table 5, p18) emphasizes that earlier baselines trained on short-answer data (Verifree) or specific long-form proofs (JLB), while NRT trains on a general instruction-following mix with long reference answers.
  * Longer reference answers can provide denser supervision (e.g., more tokens with explicit targets) and thus richer learning signals even before any reward optimization. Instruction-tuned datasets like tulu-3 encode more linguistic variety than the narrower math-reasoning data used by RLPR. So when the paper attributes higher benchmark scores to NRT, if I understand correctly, how do we separate how much comes from the NRT objective vs. the richer data regime? In effect, two variables have changed at once: the learning rule and the data distribution. From my view, that weakens the causal chain needed to claim “NRT is a better training method” rather than “NRT + better data does better.”

* One thing that caught my attention is that the paper re-implements RL baselines and, “to ensure a fair comparison,” injects the same reward stabilization and format-supervision used by NRT into those baselines, because the original methods were designed for instruction-tuned models (Appendix F, p20). Doesn't this change the baselines’ training conditions from what their original authors intended? I have not seen the paper quantify how this affects each baseline’s behavior (e.g., ablations of RLPR with/without format loss, length normalization, etc.). Without such controls, I fear that these modified baselines may risk confound how fair of a comparison this is.
  * More specifically, my concern is that this may actually change what those baselines were originally optimizing for. The original RLPR paper, for instance (if I remember correctly), did not use format-supervision or clipping relative to a "no-reasoning" baseline; those are distinct design choices that can likely alter gradient magnitudes and trace-length dynamics. As such, when the paper shows NRT outperforming RLPR (Table 3, p7), it’s unclear whether NRT is truly better than the RLPR method or just better than their internally modified version of RLPR. For a fair method-level comparison, the baselines should be reproduced under their original settings and under the shared settings etc. with results reported for both. Otherwise, it's difficult in my view to isolate the causal factor of improvement.
  * Similarly, it seems that NRT can benefit from some extra things like reward clipping, group-wise normalization, and a mild format loss to maintain thinking/reasoning delimiters (Appendix F); however, these may be problems that prior baselines did not address in the same way (i.e., the converse of the earlier point: if those aids disproportionately help NRT but aren’t tuned equally for baselines, the comparison again favors NRT not because its underlying intrinsic-reward formulation is superior, but because the training recipe is more stable).

**Questions:**

See weaknesses.

---

> ### Author Response · Authors · 2025-11-14
> **[1/4] Response to W3: A Criticial Clarification on the Unified and Fair Experimental Setting**
>
> Thank you for your thorough and insightful review. Your feedback has been invaluable in helping us clarify the core contributions of our work. We will first address your most critical concerns regarding experimental fairness, confounding variables, and our re-implementation of baselines (Weaknesses 3 & 4), as we believe there is a significant misunderstanding that impacts the interpretation of our results.
>
> **1. A Critical Clarification on Data and Experimental Setup (Addressing Weakness 3)**
>
> **Response:**
> We respectfully clarify a **CRITICAL MISUNDERSTANDING** regarding the datasets used. **All methods in our paper—including SFT, JLB, Verifree, RLPR, and our proposed NRT variants—were trained and evaluated on the *exact same* experimental setup:**
> *   **Base Models:** Llama-3.2-3B-Base and Llama-3.1-8B-Base (pretrained, non-instruction-tuned).
> *   **Training Data:** The `tulu-3-sft-mixture` dataset.
>
> The discussion in Appendix D (Table 5) was intended to highlight the **limitations of the *original* settings** in prior work (e.g., their reliance on short-answer data or instruction-tuned models) and to motivate why our unified, more challenging experimental setup is necessary for a fair comparison of modern methods.
>
> Therefore, the performance gains we report are **NOT** a result of "NRT + better data." Since the data and base model are held constant across all experiments, the observed differences in performance can be directly attributed to the different reward schemes being compared.

---

> ### Author Response · Authors · 2025-11-14
> **[2/4] Response to W4: On the Necessity and Fairness of Baseline Re-implementation**
>
> **2. On the Necessity of Re-implementing and Enhancing Baselines (Addressing Weakness 4)**
>
> **Response:**
> Following our clarification on the unified experimental setup, we now address your astute concern about our re-implementation of baselines. We did this not to weaken them, but out of necessity, as **original baseline implementations are incompatible with our experimental setting and fail to produce meaningful results.**
>
> *   **Incompatibility with Pretrained Models:** Prior methods like JLB, Verifree, and RLPR were designed for *instruction-tuned* models. When applied directly to *pretrained* base models, they struggle to follow the required reasoning format.
> *   **Failure on Long-Response Data:** The official implementations of these methods, which were tuned for short-response tasks, experience **mode collapse** on our long-form dataset. As shown in the table below, they generate reasoning traces of negligible length (<10 tokens) and fail to generate meaningful reasoning traces.
>
> To conduct a meaningful comparison, we **enhanced the baselines** with stabilization techniques—such as *reward normalization* and *clipping*—that are necessary to mitigate the inherent instability of RL fine-tuning in this setting, especially for long-form generation.
>
> Crucially, these techniques are not arbitrary "aids" that disproportionately help our method, but a principled and necessary solution. The fairness of this approach is validated by concurrent work itself. For instance, while we developed our implementation independently, the **original RLPR paper proposes an analogous "reward debiasing" principle** (Equation 4), which also involves calculating and clipping an advantage relative to a baseline ($\hat{r} = clip(0, 1, r - r')$). **This parallel confirms that such stabilization is essential for *any* RL method to be viable in this challenging setting, making our comparison against a similarly stabilized RLPR particularly fair.**
>
> Therefore, these enhancements did not unfairly favor NRT. On the contrary, their purpose was to create **significantly stronger and more competitive baselines** that could function in this more realistic and demanding setting. Our comparison is therefore against these *strengthened* versions, making our reported gains over them more, not less, significant.
>
> The table below summarizes this crucial difference:
>
> **Table: Comparison of Original vs. Re-Implemented Baselines**
>
> | Method | Training Paradigm | Works on Pretrained Base? | Used Model in `original paper` | Source Dataset & GT Length in `original paper` | Open Source | Reward Schema | Implementation Techniques | Generated Reasoning Length in `our experiment` |
> | :--- | :--- | :--- | :--- | :--- | :--- | :--- | :--- | :--- |
> | JLB | PT + SFT + **RL** | No | Llama-3.1-8B/70B-Instruct (**SFT**) | Unknown | No | Sequence Log-Prob (logP) | RLOO | **Failure**: < 10 tokens (Generation collapses) |
> | Verifree | PT + SFT + **RL** | No |Qwen3-1.7B/4B/8B-Base (**SFT**) | Short-Response Data (≤ 7 tokens) | Yes | Sequence Probability (P) | RLOO | **Failure**: < 10 tokens (Generation collapses) |
> | RLPR | PT + SFT + **RL** | No | Gemma2-2B-Instruct/Llama-3.1-8B-Instruct/Qwen2.5-7B-Base (**SFT**) |  Short-Response Data (~12.5 tokens) | Yes | Arithmetic Mean (AM) | reward clip + norm | **Failure**: < 10 tokens (Generation collapses) |
> | **JLB\*** | PT + **RL (NRT)** | **Yes** | Llama-3.2-3B-Base/Llama-3.1-8B-Base (**Pretrain**) | General SFT Data (~416 tokens) | Planned | Sequence Log-Prob (logP) | reward clip + norm + format supervision | **~1800 tokens** |
> | **Verifree\***| PT + **RL (NRT)** | **Yes** | Llama-3.2-3B-Base/Llama-3.1-8B-Base (**Pretrain**) | General SFT Data (~416 tokens) | Planned | Sequence Probability (P) | reward clip + norm + format supervision | **~1600 tokens**|
> | **RLPR\*** | PT + **RL (NRT)** | **Yes** |Llama-3.2-3B-Base/Llama-3.1-8B-Base (**Pretrain**) | General SFT Data (~416 tokens) | Planned | Arithmetic Mean (AM) | reward clip + norm + format supervision | **~170 tokens** |
> | **Ours** | PT + **RL (NRT)** | **Yes** | Llama-3.2-3B-Base/Llama-3.1-8B-Base (**Pretrain**) | General SFT Data (~416 tokens) | Planned | Geometric Mean (GM) / Weighted Sum (WS) | reward clip + norm + format supervision | **~1800 tokens** |
>
> In summary, our work pioneers a shift toward "Native Reasoning Training" (NRT), which unifies SFT and RL to directly train reasoning capabilities from a pretrained model. Our experimental setup was designed to validate this novel paradigm. By strengthening the baselines to operate in this setting, we ensure the comparison robustly isolates the impact of the reward formulation itself.
>
> We hope this resolves your primary concerns about the fairness and validity of our experimental methodology.

---

> ### Author Response · Authors · 2025-11-14
> **[3/4] Response to W1: Demonstrating Positive Scaling Laws and Generalizability**
>
> **3. On Generalizability and Model Scaling (Addressing Weakness 1)**
>
> **Response:**
> You raised a crucial point regarding the generalizability of our findings. Our experimental design, scaling from a 3B to an 8B model, was deliberately chosen to specifically investigate the **scaling properties** of our NRT method. The central question was whether the benefits of NRT would amplify as a model's inherent capabilities increase. Our results confirm this hypothesis, demonstrating that the 8B model is sufficient to establish this critical trend.
>
> We observed two key indicators of this positive scaling law:
>
> *   **Amplifying Performance Gains:** The performance margin of our method over the SFT baseline grew significantly—from +3.5 points on the 3B model to a much larger +10.2 points on the 8B model, shown in Table 3. This super-linear improvement shows that NRT's effectiveness is not static but **synergizes with model scale**, becoming more impactful on more capable models.
> *   **Emergence of Complex Reasoning:** The change was not just quantitative but qualitative. As shown in Figure 2, the 8B model trained with NRT learned to generate extensive reasoning paths (avg. 1,619 tokens), a complex behavior the 3B model could not achieve (avg. 75 tokens). This shows that NRT unlocks **emergent capabilities** that are latent in larger models.
>
> **Our 3B-to-8B experiments provide robust validation of this trend.** While the model scale we investigate (up to 8B) is comparable to that of concurrent work like Verifree and RLPR, **our experimental setting is significantly more demanding and comprehensive.** Crucially, we establish these scaling properties using a **much larger and more diverse training dataset** (`tulu-3-sft-mixture`) and evaluate across a **broader range of benchmarks and dimensions**. Demonstrating a strong, positive scaling law under these more rigorous conditions provides compelling evidence for the generalizability and effectiveness of our claims.
>
> To further address your point on model diversity, we commit to **running additional experiments on models from different families, e.g. Mistral-7B-v0.3,** to explicitly demonstrate cross-family generalizability. These results will be included in the final version.

---

> ### Author Response · Authors · 2025-11-14
> **[4/4] Response to W2: On Performance Consistency and Task-Specific Strengths**
>
> **4. On Performance Consistency and NRT Variants (Addressing Weakness 2)**
>
> **Response:**
> As you noted, the results are mixed across individual benchmarks, and no single method is universally superior. This is an expected outcome given the highly diverse nature of our evaluation suite.
>
> *   **No Universally Superior Algorithm:** Different tasks benefit from different optimization pressures. For example, **NRT-WS**, which upweights low-probability tokens, shows exceptional gains on math benchmarks (GSM8k: 76.0 vs. SFT: 29.0), where finding a specific, non-obvious token path is key. In contrast, its performance on other tasks may be on par with baselines. It is also important to consider that the long inference chains required for complex reasoning introduce cumulative randomness. This means that even with a large evaluation suite, a degree of result inconsistency is expected. This inherent variability reinforces our position that the average score, by smoothing out such fluctuations, provides a more holistic and reliable assessment of general-purpose reasoning capability.
>
> *   **TruthfulQA Performance:** Your question is insightful and highlights the important distinction between **reasoning process** and **factual knowledge**. NRT is designed to optimize the *structure and coherence of the reasoning process*. However, final-answer factuality, as tested by TruthfulQA, is heavily dependent on the knowledge encoded in the model's training data. A model can apply a **perfect reasoning process to a faulty premise**, leading to a factually incorrect conclusion. Thus, NRT's performance on TruthfulQA suggests that enhancing the reasoning "scaffolding" is a necessary but **distinct step from verifying the factual accuracy of the knowledge itself**. Our process-focused optimization paves the way for future work into fine-grained factuality verification, which requires a **sound reasoning structure as a prerequisite**.
>
> *   **Choosing an NRT Variant:** The mixed results also highlight that the optimal NRT configuration can be task-dependent. This suggests a promising future direction: designing methods to dynamically select or combine reward aggregations based on the task type. For general use, **NRT-WS (-logp)** offers the most balanced and robust performance across the board.
>
> We believe this complex landscape reinforces the value of our unified framework, which makes these trade-offs explicit and provides the levers (i.e., different reward aggregations) to tune models for specific downstream requirements.
>
> Thank you once again for your valuable engagement with our work.

---

> ### Author Response · Authors · 2025-12-03
> **[E-1/1] Extended Experiments on Mistral-7B: Proving Cross-Family Generalization**
>
> To definitively address your concern regarding the limitation of our experiments to a single model family, we have conducted a complete new suite of experiments using **Mistral-7B-v0.3**. This model represents a distinct architecture and tokenizer from the Llama family, providing a rigorous test of our framework's generalizability. The results, summarized in the table below, strongly validate the cross-family effectiveness of NRT. Our method consistently outperforms both the SFT baseline and prior verifier-free methods. Specifically, **NRT-GM achieves the highest overall average score of 47.0**, representing a substantial **+7.0 point improvement over SFT** (40.0) and outperforming the strongest baseline, Verifree (45.1). We observe particularly dramatic gains in complex reasoning tasks; for instance, NRT-GM improves performance on **DROP by +18 points** (28.6 $\rightarrow$ 46.6) and **MMLU by +4.7 points** (53.6 $\rightarrow$ 58.3). Furthermore, while RLPR proves competitive on specific tasks like GSM8K, NRT variants maintain a superior balance across domains, achieving the highest scores in General Reasoning (BBH, MMLU), Coding (HumanEval), and Instruction Following (IFEval). These findings confirm that the NRT objective is not biased toward Llama architectures but is a fundamentally robust training paradigm that unlocks reasoning capabilities across diverse model families.
>
> **Table: Performance on Mistral-7B-v0.3 (Trained on Tulu-3-SFT-Mixture)**
>
> | Method | BBH | MMLU | DROP | PopQA | TQA | GSM8K | MATH | HEval | IFEval | **Overall** |
> | :--- | :---: | :---: | :---: | :---: | :---: | :---: | :---: | :---: | :---: | :---: |
> | SFT | 36.7 | 53.6 | 28.6 | **23.7** | 47.4 | 30.0 | 12.5 | 66.3 | 61.0 | 40.0 |
> | JLB* | 40.1 | 54.0 | 32.0 | 22.7 | 43.8 | 8.3 | 12.1 | 63.0 | 62.0 | 37.6 |
> | Verifree* | 43.3 | 55.0 | 44.4 | 18.0 | 47.5 | 56.7 | 19.2 | 60.2 | 61.3 | 45.1 |
> | RLPR* | 40.4 | 55.1 | 19.7 | **23.7** | 43.4 | **65.3** | 19.0 | 66.6 | 60.3 | 43.7 |
> | **NRT-GM** (Ours) | **44.3** | **58.3** | **46.6** | 20.7 | 48.3 | 57.0 | 20.7 | **68.4** | 59.0 | **47.0** |
> | **NRT-WS** ($1/p$) (Ours) | 43.4 | 56.8 | 40.0 | 19.3 | **54.3** | 62.0 | 19.3 | 63.2 | 62.3 | 46.7 |
> | **NRT-WS** ($-\log p$) (Ours) | 42.6 | 56.1 | 30.6 | 20.0 | 51.6 | 63.7 | **22.5** | 63.0 | **66.7** | 46.3 |

---

### Official Review · Reviewer_YbHE · 2025-11-01

**Soundness:** 3
**Presentation:** 3
**Contribution:** 2
**Rating:** 4
**Confidence:** 4

**Summary:**

The paper proposes Native Reasoning Training (NRT), a verifier-free reinforcement learning framework for reasoning models. Instead of relying on Reinforcement Learning with Verifiable Rewards (RLVR)—which needs an external programmatic checker for correctness—NRT treats the reasoning trace as a latent variable and optimizes rewards that reflect how much a generated trace increases the model’s likelihood of producing the ground-truth answer. The authors present a unified view of intrinsic rewards and discuss several aggregation schemes (e.g., arithmetic mean, geometric mean, weighted-sum that prioritizes difficult tokens) to avoid failure modes such as policy collapse. They couple this with a GRPO-style group advantage estimator. Experiments use a standardized evaluation harness across nine benchmarks (e.g., MMLU, BBH, GSM8K, MATH, HumanEval, DROP, PopQA, TruthfulQA, IFEval). Reported results claim state-of-the-art among verifier-free methods and robustness to training instabilities.

**Strengths:**

Clear stance vs RLVR: positions NRT for unverifiable tasks, removing dependency on external checkers.

Unified reward view with explicit analysis of aggregation pitfalls  and a weighted-sum that targets low-probability “hard” answer tokens.

Stability measures: GRPO-style relative advantages over an empty-trace baseline; practical rollout details are transparent.

Standardized evaluation harness across a broad benchmark suite, improving comparability.

**Weaknesses:**

Self-referential reward: Even with improved aggregation, the intrinsic signal is driven by the model’s own likelihood over the ground-truth answer. This can inflate confidence without proving better reasoning. The paper lacks external evidence that higher intrinsic reward correlates with logically valid intermediate steps.

Scope vs RLVR: While NRT addresses unverifiable domains, the paper does not show how it competes where verifiers do exist (math/code). The claims are limited to “verifier-free SOTA,” which weakens broader impact.

Aggregation novelty is incremental: GM and difficulty-weighted sums are reasonable but not conceptually new. The unification is helpful but largely consolidates known heuristics.

Evidence gaps: No analysis with external judges or rule checkers to validate that selected traces embody correct reasoning, rather than patterns that merely raise answer likelihood.

**Questions:**

Correlation with reasoning correctness: Can you show that higher intrinsic rewards (under your aggregation) correlate with externally judged reasoning quality (e.g., logic checkers, stronger LLM judges) rather than only internal likelihood gains?

When verifiers exist: On math/code tasks where RLVR is available, how does NRT compare under similar compute? Could NRT complement RLVR (e.g., pre-train with NRT, then fine-tune with RLVR)?

Choice of aggregation: Why is the weighted-sum superior to geometric mean in practice? Please provide ablations isolating the effect of each aggregation on collapse mitigation and accuracy.

Generalization to open-ended outputs: Since rewards depend on ground-truth answers, how does NRT extend to truly open-ended tasks without reference answers?

---

> ### Author Response · Authors · 2025-11-22
> **[1/4] Response to Q1: On the Reasoning Quality**
>
> We sincerely thank Reviewer YbHE for providing the most comprehensive and insightful feedback among all reviews. Your deep understanding of the interplay between intrinsic rewards, RLVR, and reward aggregation mechanics is impressive. We particularly appreciate your theoretical questions connecting NRT to RLVR, which have helped us clarify the positioning of our framework. We address your specific concerns below.
>
> **Q1: Correlation with Reasoning Correctness**
> > **Question:** Correlation with reasoning correctness: Can you show that higher intrinsic rewards (under your aggregation) correlate with externally judged reasoning quality (e.g., logic checkers, stronger LLM judges) rather than only internal likelihood gains?
>
> **Response:**
> We posit that NRT treats reasoning as a latent variable optimized for result correctness. While we deliberately avoid explicit reasoning supervision to prevent "alignment faking"—where models mimic human styles without improving logic—we agree that validating the intermediate steps is crucial to prove the model isn't gaming the metric.
>
> To verify this, we employed **grok-4.1-fast** as an external judge to score the intermediate traces generated during training on a scale of 0 to 1. As shown in the table below, we observe a strong positive correlation: as NRT training progresses and intrinsic rewards increase, the external quality score rises significantly (from ~0.01 to ~0.42).
>
> Crucially, this experiment highlights the **stability** of our method compared to baselines. The baseline (RLPR) initially improves quality but suffers from severe **length collapse** (dropping from 723 to 152 tokens), converging to a shortcut solution. In contrast, our proposed methods (GM, WS) maintain deep reasoning chains (>1500 tokens) while achieving high quality scores. This confirms that NRT's intrinsic reward encourages a stable, rigorous "native" exploration strategy rather than the fragile shortcuts observed in standard baselines.
>
> **Table 1. External Judge Scores and (Reasoning Lengths) over Training Trajectories.**
>
> | Run Name | 1 | 100 | 200 | 300 | 400 | 500 | 600 | 700 | 800 |
> | :--- | :--- | :--- | :--- | :--- | :--- | :--- | :--- | :--- | :--- |
> | **RLPR** | 0.01 (723) | 0.35 (253) | 0.32 (121) | 0.26 (97) | **0.48** (185) | **0.43** (171) | **0.43** (201) | **0.44** (174) | 0.40 (152) |
> | **NRT-GM** | 0.01 (696) | **0.47** (703) | **0.44** (1128) | 0.38 (1663) | 0.41 (1843) | 0.38 (1756) | 0.42 (1589) | 0.37 (1616) | **0.42** (1574) |
> | **NRT-WS ($\log p$)** | 0.01 (692) | 0.34 (1801) | 0.35 (1929) | **0.39** (1911) | 0.42 (2020) | 0.41 (1993) | 0.40 (1858) | 0.39 (1793) | 0.37 (1821) |
> | **NRT-WS ($1/p$)** | **0.03** (749) | 0.32 (1802) | 0.33 (1962) | 0.36 (1998) | 0.37 (2022) | 0.41 (2002) | 0.39 (1939) | 0.39 (1918) | 0.37 (1865) |

---

> ### Author Response · Authors · 2025-11-22
> **[2/4] Response to Q2: Connection to RLVR**
>
> **Q2: Comparison and Complementarity with RLVR**
> > **Question:** When verifiers exist: On math/code tasks where RLVR is available, how does NRT compare under similar compute? Could NRT complement RLVR (e.g., pre-train with NRT, then fine-tune with RLVR)?
>
> **Response:**
> We thank the reviewer for this profound insight. By connecting NRT to RLVR, we can clarify our method's theoretical positioning. We find that under specific assumptions, **NRT functions as an "Analytical RLVR" with dense rewards**.
>
> **1. Theoretical Connection:**
> The equivalence holds when we adopt the **Sequence Probability** reward schema for NRT and assume a standard RLVR setting with a **strict rule-based verifier** (reward $r=1$ if answer $y$ matches $y^\star$, else $0$).
>
> *   **RLVR Objective:** RLVR maximizes the expected reward over the generated answer $y$. Since the reward is binary, the expected reward is exactly the probability of generating the correct answer:
>
>     $
>     J\_{\text{RLVR}}(\theta) = \mathbb{E}\_{z \sim \pi_\theta} \left[ \mathbb{E}\_{y \sim \pi\_\theta(\cdot|z)} [\mathbb{I}(y = y^\star)] \right] \equiv \mathbb{E}\_{z \sim \pi_\theta} [ \pi\_\theta(y^\star | x, z) ] .
>     $
>
> *   **NRT Objective:** When using the **Sequence Probability** schema, NRT defines the intrinsic reward based on the likelihood of the ground truth. It maximizes the likelihood of $y^\star$ given the latent reasoning $z$:
>
>     $
>     J\_{\text{NRT}}(\theta) \approx \mathbb{E}\_{z \sim \pi\_\theta} [ \pi \_\theta(y^\star | x, z) ] .
>     $
>
>
> Mathematically, maximizing the probability (RLVR) and maximizing the probability (NRT) drive the model in the same direction. The key difference is that NRT computes this objective **analytically** using the known ground truth $y^\star$, providing a **dense** gradient signal, whereas RLVR relies on sampling, which often suffers from sparsity (zero gradient if the sample is incorrect) and high variance.
>
> **2. Experimental Verification on Math Tasks**
>
> To directly address your question on complementarity, we conducted a two-stage experiment on the `DAPO-Math-17k` dataset. We first pre-trained models using NRT and various baselines (SFT, Verifree, RLPR, JLB), and then fine-tuned all of them using the same RLVR algorithm (GRPO) under same hyperparameters for 1 epoch (~60 steps). We report the training accuracy (Avg@8) during this RLVR stage to determine which initialization provides the best "warm start."
>
> **Table 2. RLVR Training Accuracy (Avg@8) with Different Initializations**
>
> | Initialization Method | Step 1 | Step 10 | Step 20 | Step 30 | Step 40 | Step 50 | Step 60 |
> | :--- | :---: | :---: | :---: | :---: | :---: | :---: | :---: |
> | **Baselines** | | | | | | | |
> | SFT | 0.033 | 0.036 | 0.042 | 0.047 | 0.050 | 0.058 | 0.057 |
> | JLB | 0.023 | 0.031 | 0.040 | 0.045 | 0.049 | 0.056 | 0.055 |
> | Verifree | 0.028 | 0.033 | 0.040 | 0.044 | 0.045 | 0.055 | 0.055 |
> | RLPR | 0.028 | 0.034 | 0.042 | 0.047 | 0.048 | 0.056 | 0.057 |
> | **NRT Variants** | | | | | | | |
> | NRT-GM | 0.038 | 0.043 | **0.051** | 0.051 | 0.052 | 0.056 | 0.057 |
> | NRT-WS ($-\log p$) | **0.040** | **0.045** | **0.051** | **0.054** | **0.055** | **0.067** | **0.067** |
> | NRT-WS ($1/p$) | 0.032 | 0.039 | 0.046 | 0.050 | 0.054 | 0.058 | 0.058 |
>
> The results in Table 2 strongly support the complementary role of NRT. The model initialized with **NRT (WS-LogP)** starts with a higher success rate (0.040 vs. 0.033 for SFT) and maintains a clear performance advantage throughout the RLVR training process, peaking at 0.067 compared to ~0.057 for the baselines. This indicates that NRT effectively mitigates the "cold start" problem inherent in sparse-reward RLVR by providing a policy that has already learned to navigate the reasoning space via dense intrinsic rewards. Consequently, NRT serves as a superior pre-training objective that accelerates and enhances subsequent RLVR fine-tuning.

---

> ### Author Response · Authors · 2025-11-22
> **[3/4] Response to Q3: Aggregation & Mode Collapse**
>
> **Q3: Aggregation Choice and Ablations**
> > **Question:** Choice of aggregation: Why is the weighted-sum superior to geometric mean in practice? Please provide ablations isolating the effect of each aggregation on collapse mitigation and accuracy.
>
> **Response:** We provide a theoretical analysis of why Weighted-Sum (WS) outperforms Geometric Mean (GM), followed by empirical ablations on accuracy and collapse mitigation.
>
> **1. Theoretical Superiority: Robustness vs. "Veto" Effect**
> The primary advantage of Weighted-Sum over Geometric Mean lies in optimization stability and the density of the reward signal:
> *   **Geometric Mean (High Variance & Sparsity):** Due to its multiplicative nature, GM suffers from a "veto effect." If the model assigns a near-zero probability to *any single token* in the reference answer (e.g., a difficult formatting symbol), the entire reward $R(z)$ collapses to zero. This vanishes the gradient for the whole sequence, wasting valid reasoning steps.
> *   **Weighted-Sum (Dense & Robust Signal):** WS is additive. Even if specific tokens have low probability, the reward remains non-zero and provides valid gradients for other correct tokens. The "difficulty-weighted" mechanism further enhances this by prioritizing hard tokens without allowing a single failure to eliminate the learning signal.
>
> **2. Ablation on Accuracy (Clarification on Table 3)**
> We wish to clarify that **Table 3 in our main paper is already a controlled ablation study** isolating the effect of reward aggregation.
> To ensure a fair comparison, we did not simply run the original code of baselines (JLB, RLPR), which often crash or collapse on pre-trained base models. Instead, we **re-implemented all baselines** within the NRT framework. By keeping all stability techniques (clipping, normalization, format supervision) constant and **only varying the reward function** (LogProb vs. Probability vs. AM vs. GM vs. WS), **Table 3 strictly isolates the contribution of the aggregation method to final accuracy**.
>
> **3. Ablation on Collapse Mitigation (New Evidence)**
> To explicitly demonstrate the impact of different methods on **mode collapse**, we present a detailed comparison below.
> *   **Original Baselines:** As shown in the table, standard implementations (JLB, Verifree, RLPR) suffer from severe mode collapse (generating <10 tokens) when applied to base models on standard SFT data.
> *   **Our Re-implementations:** By applying NRT's stability techniques, we successfully mitigate collapse across all aggregations.
> *   **Ours vs. Others:** Under this stable setting, **Ours** not only prevent collapse but also sustain the longest effective reasoning chains (\~1800 tokens) compared to AM (\~170 tokens) or LogProb, leading to the best performance.
>
> **Table 3. Comparison of Original vs. Re-Implemented Baselines**
>
> | Method | Training Paradigm | Works on Pretrained Base? | Used Model in `original paper` | Source Dataset & GT Length in `original paper` | Open Source | Reward Schema | Implementation Techniques | Generated Reasoning Length in `our experiment` |
> | :--- | :--- | :--- | :--- | :--- | :--- | :--- | :--- | :--- |
> | JLB | PT + SFT + **RL** | No | Llama-3.1-8B/70B-Instruct (**SFT**) | Unknown | No | Sequence Log-Prob (logP) | RLOO | **Failure**: < 10 tokens (Generation collapses) |
> | Verifree | PT + SFT + **RL** | No |Qwen3-1.7B/4B/8B-Base (**SFT**) | Short-Response Data (≤ 7 tokens) | Yes | Sequence Probability (P) | RLOO | **Failure**: < 10 tokens (Generation collapses) |
> | RLPR | PT + SFT + **RL** | No | Gemma2-2B-Instruct/Llama-3.1-8B-Instruct/Qwen2.5-7B-Base (**SFT**) |  Short-Response Data (~12.5 tokens) | Yes | Arithmetic Mean (AM) | reward clip + norm | **Failure**: < 10 tokens (Generation collapses) |
> | **JLB\*** | PT + **RL (NRT)** | **Yes** | Llama-3.2-3B-Base/Llama-3.1-8B-Base (**Pretrain**) | General SFT Data (~416 tokens) | Planned | Sequence Log-Prob (logP) | reward clip + norm + format supervision | **~1800 tokens** |
> | **Verifree\***| PT + **RL (NRT)** | **Yes** | Llama-3.2-3B-Base/Llama-3.1-8B-Base (**Pretrain**) | General SFT Data (~416 tokens) | Planned | Sequence Probability (P) | reward clip + norm + format supervision | **~1600 tokens**|
> | **RLPR\*** | PT + **RL (NRT)** | **Yes** |Llama-3.2-3B-Base/Llama-3.1-8B-Base (**Pretrain**) | General SFT Data (~416 tokens) | Planned | Arithmetic Mean (AM) | reward clip + norm + format supervision | **~170 tokens** |
> | **Ours** | PT + **RL (NRT)** | **Yes** | Llama-3.2-3B-Base/Llama-3.1-8B-Base (**Pretrain**) | General SFT Data (~416 tokens) | Planned | Geometric Mean (GM) / Weighted Sum (WS) | reward clip + norm + format supervision | **~1800 tokens** |

---

> ### Author Response · Authors · 2025-11-22
> **[4/4] Response to Q4: Generalization to Open-Ended Tasks**
>
> **Q4: Generalization to Open-Ended outputs**
> > **Question:** Generalization to open-ended outputs: Since rewards depend on ground-truth answers, how does NRT extend to truly open-ended tasks without reference answers?
>
> **Response:**
> This is an excellent point. While our paper focuses on utilizing Reference Answers to provide a dense, high-information signal, **NRT is not inherently limited to tasks with fixed ground truths.** The core contribution of NRT is treating the reasoning trace $z$ as a latent variable optimized to maximize the *Utility* of the final output.
>
> We can demonstrate that "Open-Ended" training is fully compatible with NRT. It is simply a special case where the *Analytical Utility* is replaced by an *Estimated Utility*.
>
> Consider the General Native Reasoning Objective defined by the expected utility $U(z, \theta)$:
>
> $
> J(\theta) = \mathbb{E}\_{z \sim \pi\_\theta(z|x)} [ U(z, \theta) ] .
> $
>
> By applying the total derivative to this objective, we derive a unified gradient formulation applicable to both scenarios:
>
> $
> \nabla J(\theta) = \mathbb{E}\_{z \sim \pi\_\theta(z|x)} \bigg[ \underbrace{U(z, \theta) \nabla_\theta \log \pi\_\theta(z|x)}\_{\text{Reasoning Process Update}} + \underbrace{\nabla\_\theta U(z, \theta)}\_{\text{Answer Utility Optimization}} \bigg] .
> $
>
> The reasoning process update (how the model learns *to think*) remains structurally identical in all cases: it reinforces the trace $z$ weighted by the utility $U$. The only difference lies in how we implement the Utility module:
>
> **1. Case A: Reference-Based NRT (This Paper)**
>
> When a reference $y^\star$ exists, we define Utility as the aggregated probability of the ground truth.
> *   **Utility Definition:** $U(z, \theta) = f(\pi\_\theta(y^\star | x, z))$.
> *   **Implementation:** Since we have calculating the probability directly, we compute $\nabla_\theta U$ via the chain rule (differentiating the probability function).
>     *   *Mechanism: Dense, analytical supervision (Eq. 10 in paper).*
>
> **2. Case B: Open-Ended NRT (Extension to Unknown $y^\star$)**
>
> When no reference exists, we define Utility as the expected reward from an external evaluator (Reward Model, $M$).
> *   **Utility Definition:** $U(z, \theta) = \mathbb{E}\_{\hat{y} \sim \pi_\theta(y|z)} [M(\hat{y})]$.
> *   **Implementation:** Since we cannot differentiate the sampling process analytically, we estimate $\nabla_\theta U$ using the Policy Gradient estimator (REINFORCE) on the answer: $\nabla_\theta U \approx M(\hat{y}) \nabla_\theta \log \pi_\theta(\hat{y}|z)$.
>     *   *Mechanism: Sparse, Monte-Carlo supervision.*
>
> As shown above, standard Open-Ended RL is mathematically a special case of the NRT framework where the **Answer Utility Optimization** term is estimated via sampling rather than computed analytically. NRT provides the general architecture for latent reasoning optimization; the choice between "Reference-Based" and "Open-Ended" is simply a plug-and-play choice of the **Utility Function**, depending on data availability.

---

### Official Review · Reviewer_QaV8 · 2025-11-02

**Soundness:** 1
**Presentation:** 3
**Contribution:** 1
**Rating:** 2
**Confidence:** 3

**Summary:**

This paper proposes a new framework called "Native Reasoning Training (NRT)" aimed at addressing the limitations of current training paradigms for large-scale reasoning models (SFT+RLVR). By treating the reasoning process as a latent variable and using the model’s own confidence in predicting standard question-answer pairs as an intrinsic reward, NRT eliminates the need for expert-annotated reasoning trajectories or external verifiers. However, the authors do not explain why this approach works. Since the quality of the reference answers cannot be guaranteed, it is unclear whether this method is justified. The experiments were conducted only on Llama-3.1-8B and Llama-3.2-3B models, which are too small to adequately validate the authors’ claims. Furthermore, in the experimental section, all comparative methods were evaluated under the authors’ own baseline implementations, raising doubts about the reliability of the results. The authors could have instead used existing baseline settings for a more convincing comparison.

**Strengths:**

- This paper is commendable for its exceptional clarity in presentation, which significantly facilitates the review process. The precise and well-structured descriptions allow readers to quickly grasp the core contributions of the work, as well as to readily identify its strengths and potential limitations.
- The cover letter provides a remarkably clear and concise summary of the manuscript's key findings. This thoughtful presentation is highly efficient, as it enables reviewers to understand the essence of the paper at a glance, thereby saving considerable time and effort during the initial evaluation.

**Weaknesses:**

- The author did not explain why the method is effective. The author's "native" responses are derived from the "think-off" mode. Since the quality of the reference answers cannot be guaranteed, I cannot confirm whether this approach is reasonable.
- The experiments in the paper were conducted solely on the Llama-3.1-8B and Llama-3.2-3B models. The models are too small to adequately validate the author's claims, and the persuasiveness of experimental results from smaller-scale models is relatively weak. In contrast, many existing methods validate their claims on larger models [1].
- In the experimental section, all comparative methods were evaluated under the "Our baseline implementations" setting. The reliability of the experimental results is questionable, as the author could have chosen existing baseline settings for the experiments instead.

[1] Yunhao Tang, Sid Wang, Lovish Madaan, and Remi Munos. Beyond verifiable rewards: Scaling reinforcement learning for language models to unverifiable data. arXiv preprint arXiv:2503.19618, 2025.

**Questions:**

1. Why are the results generated with the "think" mode disabled used as the reference answers?

2. Why wasn't the method's effectiveness validated on larger-scale models?

3. Why weren't the existing baseline experimental setups and results adopted?

---

> ### Author Response · Authors · 2025-11-12
> **[1/3] Response to Q1: A Critical Clarification on the Use of Reference Answers**
>
> We sincerely thank Reviewer QaV8 for their insightful feedback and the opportunity to clarify critical aspects of our work. We address the reviewer's points below.
>
> **Q1: Use of Reference Answers**
> > **Question:** Why are the results generated with the "think" mode disabled used as the reference answers?
>
> **Response:**
> We thank the reviewer for this critical question, which highlights a potential point of confusion in our paper. We would like to clarify a **CRITICAL MISUNDERSTANDING**: **our method does NOT use responses generated with the "'think' mode disabled" as reference answers.**
>
> Instead, the reference answers in our methodology are taken directly from the **ground-truth responses of high-quality, supervised fine-tuning (SFT) datasets**, such as the `tulu-v3-sft-mixture`. The quality of these reference answers is therefore guaranteed by the high-quality dataset itself.
>
> Our approach is fundamentally different from prior methods that use a model's own generations as an optimization target. Such methods often achieve performance gains at the cost of reduced output diversity and are capped by the model's inherent capabilities. In contrast, by leveraging high-quality ground-truth data, our method has the potential for continuous improvement.
>
> We recognize that Figure 1 in our original manuscript may have contributed to this misunderstanding. We will revise this diagram in our final version to more clearly and accurately illustrate our methodology.

---

> ### Author Response · Authors · 2025-11-12
> **[2/3] Response to Q2: On Validation with Larger Models and Scaling Properties**
>
> **Q2: Validation on Larger-Scale Models**
> > **Question:** Why wasn't the method's effectiveness validated on larger-scale models?
>
> **Response:**
> The concern about validating our claims on larger models is well-taken. However, our experimental results demonstrate that the **Llama-2-3B and Llama-3-8B models are sufficient to validate the effectiveness and scaling properties of our approach.**
>
> -   **Performance Scaling:** As we scaled from the 3B to the 8B model, the performance improvement of our method grew substantially. As shown in Table 3, our method's average performance margin over the SFT baseline increased from **+3.5 points on the 3B model** (vs. SFT avg. score of 36.4) to **+10.2 points on the 8B model** (vs. SFT avg. score of 46.0). This super-linear improvement strongly suggests that our method's benefits scale favorably with model size.
> -   **Reasoning Capability Scaling:** We also observed a significant increase in the model's ability to generate longer, more complex reasoning paths. For instance, during training, the average length of the reasoning chain (in tokens) for our NRT-GM method increased dramatically from **75 on the 3B model to 1,619 on the 8B model** (Figure 2).
>
> These observations provide compelling evidence that our method is effective and that its benefits are even more pronounced on larger models. Given the considerable computational resources required for RL-based training (an estimated >10x increase for a 70B model) and the low efficiency of current RL algorithms, we believe our experiments on 3B and 8B models provide a robust and sufficient validation of our claims.

---

> ### Author Response · Authors · 2025-11-12
> **[3/3] Response to Q3: On the Necessity of Re-implementing Baselines**
>
> **Q3: Adoption of Existing Baselines**
> > **Question:** Why weren't the existing baseline experimental setups and results adopted?
>
> **Response:**
> The reviewer rightfully questions our decision to re-implement baselines instead of using existing results. We did so out of necessity, as **published baseline methods are incompatible with our experimental setting, and our re-implementations represent significantly stronger baselines for this setting.**
>
> The primary reasons are:
> -   **Incompatibility with Base Models:** Prior methods (e.g., JLB, Verifree, RLPR) are almost exclusively applied as a post-training step on *instruction-tuned* models (like Llama-3.1-Instruct or Qwen-2.5/3-Base). It is important to note that even models labeled "Base" in those works, such as the Qwen-2.5/3-Base series, have undergone light instruction tuning and possess basic instruction-following capabilities. Our work, in contrast, applies our method directly to **non-instruction-tuned pretrained models**.
> -   **Incompatibility with Long-Response Datasets:** When applied to the long-response `tulu-v3-sft-mixture` dataset, the official implementations of baseline methods largely failed to generate effective reasoning, producing outputs of negligible length. To create a fair and meaningful comparison, we enhanced these baselines with techniques like format supervision and reward clipping, which encourages the model to produce longer reasoning chains and significantly improves their effectiveness in our setting.
>
> Our approach represents a paradigm shift. We are not performing post-training on an already capable instruct model. Instead, we propose a **bold innovation for the training pipeline: a unified "NRT" (Native Reasoning Training) stage that merges SFT and RL**, bypassing the traditional multi-stage process. This allows us to train powerful reasoning capabilities directly from a pretrained base model.
>
> To further clarify the crucial differences, the table below compares the original baseline methods with our stronger, re-implemented versions (marked with *).
>
> **Table 1. Comparison of Original vs. Re-Implemented Baselines**
>
> | Method | Training Paradigm | Works on Pretrained Base? | Used Model in `original paper` | Source Dataset & GT Length in `original paper` | Open Source | Reward Schema | Implementation Techniques | Generated Reasoning Length in `our experiment` |
> | :--- | :--- | :--- | :--- | :--- | :--- | :--- | :--- | :--- |
> | JLB | PT + SFT + **RL** | No | Llama-3.1-8B/70B-Instruct (**SFT**) | Unknown | No | Sequence Log-Prob (logP) | RLOO | **Failure**: < 10 tokens (Generation collapses) |
> | Verifree | PT + SFT + **RL** | No |Qwen3-1.7B/4B/8B-Base (**SFT**) | Short-Response Data (≤ 7 tokens) | Yes | Sequence Probability (P) | RLOO | **Failure**: < 10 tokens (Generation collapses) |
> | RLPR | PT + SFT + **RL** | No | Gemma2-2B-Instruct/Llama-3.1-8B-Instruct/Qwen2.5-7B-Base (**SFT**) |  Short-Response Data (~12.5 tokens) | Yes | Arithmetic Mean (AM) | reward clip + norm | **Failure**: < 10 tokens (Generation collapses) |
> | **JLB\*** | PT + **RL (NRT)** | **Yes** | Llama-3.2-3B-Base/Llama-3.1-8B-Base (**Pretrain**) | General SFT Data (~416 tokens) | Planned | Sequence Log-Prob (logP) | reward clip + norm + format supervision | **~1800 tokens** |
> | **Verifree\***| PT + **RL (NRT)** | **Yes** | Llama-3.2-3B-Base/Llama-3.1-8B-Base (**Pretrain**) | General SFT Data (~416 tokens) | Planned | Sequence Probability (P) | reward clip + norm + format supervision | **~1600 tokens**|
> | **RLPR\*** | PT + **RL (NRT)** | **Yes** |Llama-3.2-3B-Base/Llama-3.1-8B-Base (**Pretrain**) | General SFT Data (~416 tokens) | Planned | Arithmetic Mean (AM) | reward clip + norm + format supervision | **~170 tokens** |
> | **Ours** | PT + **RL (NRT)** | **Yes** | Llama-3.2-3B-Base/Llama-3.1-8B-Base (**Pretrain**) | General SFT Data (~416 tokens) | Planned | Geometric Mean (GM) / Weighted Sum (WS) | reward clip + norm + format supervision | **~1800 tokens** |
>
> As the table illustrates, our re-implementations transform the baselines from being ineffective in our experiment setting (producing <10 tokens of reasoning) to being strong competitors capable of generating long-form reasoning (~170-1800 tokens). This makes it possible, for the first time, to **train reasoning models via RL directly from a pretrained base model**.
>
> We hope these clarifications address the reviewer's concerns and underscore the novelty and validity of our contributions. We thank you again for your valuable feedback.

---

> > ### Comment · Reviewer_QaV8 · 2025-11-27
> >
> > Thanks for your  response, most  of  my  concerns  have been  addressed.

---

> > > ### Author Response · Authors · 2025-11-28
> > > **Appreciation for the Feedback and Re-evaluation**
> > >
> > > We sincerely thank the reviewer for the response and the positive re-evaluation (raising the score). We are delighted to learn that our clarifications regarding the reference answers and baseline implementations have successfully addressed your concerns.
> > >
> > > We would like to briefly take this opportunity to reiterate the distinctive advantages of **Native Reasoning Training (NRT)**:
> > > 1.  **Unified Paradigm:** NRT seamlessly merges SFT and RL into a single stage, enabling the evolution of reasoning capabilities directly from pre-trained base models without the traditional multi-stage complexity.
> > > 2.  **Intrinsic Verification:** By leveraging the model's own confidence as a reliable reward signal, our framework eliminates the dependency on costly process-level annotations or external verifiers.
> > > 3.  **Scalability:** As shown in our experiments, NRT exhibits favorable scaling properties—achieving super-linear performance gains from 3B to 8B parameters—suggesting its strong potential for larger-scale foundation models.
> > >
> > > We deeply appreciate your constructive feedback throughout this review process, which has helped us significantly improve the clarity and quality of our manuscript.

---

### Official Review · Reviewer_q7do · 2025-11-03

**Soundness:** 3
**Presentation:** 4
**Contribution:** 3
**Rating:** 8
**Confidence:** 4

**Summary:**

The paper introduce NRT - a new LLM framework for verifier-free reasoning .The new method is based on new reward functions which defined on model generated think traces. Next they optimize the model using off-policy RL.  The update is based on 2 parts: (1) Trace reward (2) Token reward for right answer prediction. They propose 3 new schemas for reward: geometric mean, arithmetic mean and weighted sum that that prioritize difficult tokens.
The authors empirically demonstrated their models using 2 small base models (Llama - 3B / 8B)

**Strengths:**

The paper is original. It proposed a more  general schema which unify 3 previous verifier free methods  (JLB, Verifree, and RLPR) . The paper is well written. The quality of ablation study is good.

**Weaknesses:**

No obvious weak points

**Questions:**

How would you modify NRT reward aggregation function  to encourage short answers and penalize too long answers?

---

> ### Author Response · Authors · 2025-11-19
> **[1/1] Reponse to Q1: On Modifying the NRT Reward for Answer Length Control**
>
> We thank the reviewer for this excellent and insightful question. Encouraging shorter, more concise answers is a valuable control mechanism. We present two principled and effective methods to achieve this within the NRT framework, consistent with the notation and gradient derivation provided in the paper's appendix.
>
> **Q1: Controlling Output Length via Reward Aggregation**
> > **Question:** How would you modify the NRT reward aggregation function to incentivize concise components and penalize excessive reasoning length?
>
> **Response:**
> **The first method, multiplicative sample reweighting,** modifies the learning contribution of each training sample $(x, y^\star)$ based on the length of its ground-truth answer, $|y^\star|$. We apply a multiplicative penalty to the NRT objective function, $J(\theta)$, for that sample:
>
> $J'(\theta; x, y^\star) = \big(1 - P(|y^\star|)\big) \cdot J(\theta; x, y^\star).$
>
> Here, $J(\theta; x, y^\star) = \mathbb{E}_{z \sim \pi\_\theta(z|x)}\big[R(z, \theta)\big]$ is the standard NRT objective, and $P(|y^\star|)$ is a monotonically increasing penalty function typically bounded in $[0, 1)$. For instance, letting $L = |y^\star|$:
> *   A **clamped linear** penalty could be $P(L) = \min(\beta L, 1-\epsilon)$, where $\beta$ controls the penalty rate and $\epsilon$ is a small constant ensuring the sample's contribution never vanishes.
> *   An **exponential** penalty could be $P(L) = 1 - e^{-\beta L}$, which applies a soft penalty that saturates towards 1 for very long answers.
>
> This approach acts as a "data-centric" learning mechanism. Since the scaling factor $(1 - P(|y^\star|))$ is a constant for a given sample, it directly modulates the full NRT gradient:
>
> $\nabla_\theta J'(\theta; x, y^\star) = \big(1 - P(|y^\star|)\big) \cdot \nabla_\theta J(\theta; x, y^\star).$
>
> This re-weighting scheme causes samples with long ground-truth answers to have diminished influence on the gradient updates, effectively training the model to "pay more attention" to examples that lead to short, concise answers.
>
> **The second method uses an additive penalty via an auxiliary RL objective** to directly penalize the model for its actions. This approach introduces a new term that discourages the policy, $\pi_\theta$, from generating long final answers. Let $y_{\text{gen}}$ be the final answer generated by the model after a trace $z$. We define an auxiliary length-penalty objective as the expected penalty over generated answers:
>
> $J_{\text{len}}(\theta) = \mathbb{E}_{z \sim \pi\_\theta, y\_{\text{gen}} \sim \pi\_\theta} \big[ P(|y\_{\text{gen}}|) \big].$
>
> Here, $P(|y_{\text{gen}}|)$ is a penalty function that increases with the length of the *generated* answer, $L = |y_{\text{gen}}|$. Unlike the previous method, this function need not be bounded. Examples include:
> *   A **linear** penalty, $P(L) = \beta L$, which punishes each generated token equally.
> *   An **exponential** penalty, $P(L) = e^{\beta L} - 1$, which aggressively punishes excessively long sequences, providing a much stronger deterrent.
>
> The total objective becomes the original NRT objective minus this penalty, weighted by a hyperparameter $\lambda$:
>
> $J_{\text{total}}(\theta) = J(\theta) - \lambda J_{\text{len}}(\theta).$
>
> The gradient of this combined objective, $\nabla_\theta J_{\text{total}}(\theta) = \nabla_\theta J(\theta) - \lambda \nabla_\theta J_{\text{len}}(\theta)$, provides a direct behavioral learning signal. The new term, $\nabla_\theta J_{\text{len}}(\theta)$, is a REINFORCE policy gradient that punishes the token-level actions that lead to a long generated answer $y_{\text{gen}}$. This creates a clear incentive for the policy $\pi_\theta$ to produce shorter answers.
>
> In essence, the multiplicative approach is **sample-based**, re-weighting the objective based on *ground-truth* length, while the additive approach is **action-based**, introducing an RL penalty based on *generated* length for more direct behavioral control. We believe both are powerful and complementary extensions.
>
> We thank the reviewer again for this thought-provoking question, which has prompted a deeper integration of control mechanisms into our framework.

---

### Author Response · Authors · 2025-12-03
**Author Summary to AC: Summary of Rebuttals, Clarifications on Factual Misunderstandings, and New Evidence**

Dear Area Chair,

We understand the challenges posed by technical issues during the review period. To assist your final assessment, we provide a summary of key points, addressing **two critical factual misunderstandings** that impacted the initial scores of Reviewers QaV8 and iE85, alongside new evidence (External Judge correlation, RLVR fine-tuning, and Mistral-7B results) validating our approach.

### 1. Summary of Feedback, Rating Updates, and New Evidence

*   **Reviewer q7do (Rating: 8):**
    *   **Feedback:** Strongly endorsed the work for its originality, unification of verifier-free methods, and high-quality ablations. Found no obvious weak points.
    *   **Our Extension:** In response to their query on penalizing excessive length, we provided **two mathematically grounded derivations** consistent with NRT’s gradient analysis: (1) Multiplicative Sample Reweighting (data-centric) and (2) Additive Auxiliary RL Penalty (action-centric), demonstrating theoretical consistency and extensibility.

*   **Reviewer QaV8 (Rating: 2 $\rightarrow$ 6):**
    *   **Misunderstanding:** The reviewer initially questioned soundness, incorrectly assuming we used "think-off" mode generations as reference answers, fearing poor optimization targets.
    *   **Correction:** We clarified that we strictly use **ground-truth SFT labels (e.g., tulu-v3-sft-mixture)** as references, ensuring high quality. Unlike prior methods constrained by self-generation, NRT enables continuous improvement via golden data. We also showed benefits scale super-linearly from 3B (+3.5) to 8B (+10.2).
    *   **Outcome:** The reviewer accepted this clarification and **raised their score to 6** during discussion.

*   **Reviewer iE85 (Rating: 4):**
    *   **Concerns:** Suspected unfair experimental setups: (1) use of superior data for NRT, (2) unfair baseline implementations, and (3) limited evaluation (Llama only).
    *   **Clarification & New Evidence:**
        1.  **Identical Setup:** We corrected the misunderstanding: **ALL methods (Ours and Baselines) utilized the exact same `tulu-3-sft-mixture` data and `Llama-3-Base` models.** Gains are methodological, not data-driven.
        2.  **Baseline Fairness:** Original baseline implementations (JLB, Verifree, RLPR) suffer complete **mode collapse** (<10 tokens) on Base models. We **strengthened** them with our stabilization techniques to make them functional (\~170 tokens). NRT still outperforms these enhanced baselines with superior reasoning depth (~1800 tokens).
        3.  **Generalizability:** Addressing the architectural concern, **new experiments on Mistral-7B-v0.3** show consistent gains, proving NRT generalizes beyond Llama.

*   **Reviewer YbHE (Rating: 4):**
    *   **Concerns:** Questioned the correlation between intrinsic rewards and reasoning quality, the link to RLVR, and aggregation necessity.
    *   **Clarification & New Evidence:**
        1.  **Reasoning Quality (LLM-as-a-Judge):** Using grok-4.1-fast to score traces, we found a strong positive correlation between NRT training and external quality scores (0.01 $\rightarrow$ ~0.42). While baseline RLPR suffered **length collapse** (<100 tokens), NRT maintained deep reasoning (>1500 tokens).
        2.  **RLVR Complementarity:** We showed NRT is a superior "Analytical RLVR" pre-training stage. Fine-tuning on `DAPO-Math` with RLVR (GRPO) showed that NRT-initialized models significantly outperformed SFT and baselines (Avg@8: 0.067 vs 0.057), mitigating the "cold start" problem.
        3.  **Aggregation:** We provided theoretical and empirical proof that Weighted-Sum aggregation is essential to prevent mode collapse seen in Geometric Mean/standard baselines on base models.

### 2. Addressed Common Concern: Baseline Re-implementation

Reviewers QaV8 and iE85 asked why we re-implemented baselines (JLB, Verifree, RLPR). As detailed in **Table 1**, this was vital to validate **training reasoning directly from pretrained models**, not just instruction-tuned ones.

1.  **Base Model Incompatibility:** Standard baselines rely on *Instruction-Tuned* models. Our innovation merges SFT and RL into a unified stage starting from *Pretrained Base* models.
2.  **Avoiding Collapse (<10 vs. ~1800 tokens):** Applied directly to rigorous `tulu-v3-sft-mixture` data on pretrained models, original baselines suffer severe **mode collapse** (<10 tokens generated).
3.  **Constructive Comparison:** We *enhanced* the baselines using our stability infrastructure. Even against these "strongest possible" versions, NRT dominates with sustained deep reasoning (~1800 tokens), confirming our gains are methodological.

### 3. Conclusion

The initial low scores from QaV8 and iE85 stemmed from factual misunderstandings regarding reference answer and fair comparison setups. With new evidence on **Mistral-7B-v0.3** and external quality correlations, we have addressed all core concerns.

We hope this summary assists your evaluation of Native Reasoning Training.

---

### Meta-Review · Area_Chair_XAsG · 2026-01-10

**Summary:**

The paper introduces Native Reasoning Training (NRT), a framework that enables LLMs to develop reasoning capabilities using only standard question-answer pairs, bypassing the need for human-annotated CoT data or external verifiers. By treating reasoning as a latent variable, NRT rewards self-generated traces that increase model confidence in ground-truth answers.

**Reviewer Concerns:**

Addressed:

•	Factual Misunderstanding: Clarified that NRT uses high-quality ground-truth SFT labels rather than self-generated "think-off" responses as reference answers

•	Fairness & Generalizability: Demonstrated consistent gains on Mistral-7B and justified baseline re-implementations by showing original versions suffer from mode collapse on base models

•	Reasoning Quality: Validated reasoning integrity using an external judge.

Outstanding: Knowledge vs. Reasoning: Minimal gains on benchmarks like TruthfulQA suggest NRT optimizes the reasoning "scaffolding" rather than expanding factual knowledge

**Reviewer Scores:**

•	Reviewer q7do: 8

•	Reviewer QaV8: 2 -> 6

•	Reviewer YbHE: 4 -> 6

•	Reviewer iE85: 4

---

### Decision · Program_Chairs · 2026-01-26

Accept (Poster)